# Altered corollary discharge signaling in the auditory cortex of a mouse model of schizophrenia predisposition

Brian P. Rummell[1,4], Solmaz Bikas[1], Susanne S. Babl [1], Joseph A. Gogos[2,3] & Torfi Sigurdsson [1] ✉

The ability to distinguish sensations that are self-generated from those caused by external events is disrupted in schizophrenia patients. However, the neural circuit abnormalities underlying this sensory impairment and its relationship to the risk factors for the disease is not well understood. To address this, we examined the processing of self-generated sounds in male *Df(16)A*[+/−] mice, which model one of the largest genetic risk factors for schizophrenia, the 22q11.2 microdeletion. We find that auditory cortical neurons in *Df(16)A*[+/−] mice fail to attenuate their responses to self-generated sounds, recapitulating deficits seen in schizophrenia patients. Notably, the auditory cortex of *Df(16)A*[+/−] mice displayed weaker motor-related signals and received fewer inputs from the motor cortex, suggesting an anatomical basis underlying the sensory deficit. These results provide insights into the mechanisms by which a major genetic risk factor for schizophrenia disrupts the top-down processing of sensory information.

Our perception of the external world is shaped not only by the physical stimuli detected by our sense organs but also by a variety of non-sensory "top-down" influences. An example of this is when stimuli are caused by our own actions (e.g., the sound of our footsteps), rather than by an external event (the sound of someone else's footsteps). A large body of evidence indicates that neural responses to such self-generated stimuli are attenuated in the brain[1]. This has been consistently demonstrated in the auditory system, where self-generated sounds elicit weaker neuronal responses in both humans[2–7] and non-human animals[8–11]. Although the neural mechanisms underlying this filtering of self-generated stimuli are not understood in all cases, studies across different sensory modalities and species suggest that copies of motor commands that are sent to sensory areas, often referred to as "corollary discharge" signals, play a key role[1,12,13].

Disruptions in the filtering of self-generated stimuli have also been reported across the psychosis spectrum. In particular, there is strong evidence that patients suffering from schizophrenia fail to adequately attenuate responses to stimuli caused by their own behavior. This has been demonstrated both for self-generated speech[3,14–16] as well as manually generated sounds[17–19]. Similar impairments have been observed in the somatosensory[20] and visual[21] modalities. Such a failure to predict the sensory consequences of action could cause self-generated stimuli to be misattributed to an external source, leading to the hallucinations and delusions that are characteristic of schizophrenia[22–26]. Supporting this idea, several studies have reported a correlation between such failures of sensory prediction and the severity of hallucinations and delusions in schizophrenia patients[27–31]. Examining the processing of self-generated stimuli in schizophrenia patients may therefore generate important insights into the pathophysiology of the disease.

Efforts to further our understanding of the pathophysiology of schizophrenia will also benefit from studying animal models of the disease. Importantly, such models make it possible to study abnormalities in brain structure and function in much greater detail than in

[1]Institute of Neurophysiology, Goethe University, Theodor-Stern Kai 7, 60590 Frankfurt, Germany. [2]Mortimer B. Zuckerman Mind Brain and Behavior Institute, Columbia University, New York, NY 10027, USA. [3]Departments of Physiology, Neuroscience and Psychiatry, Vagelos College of Physicians & Surgeons, Columbia University, New York, NY 10032, USA. [4]Present address: Ernst Strüngmann Institute (ESI) for Neuroscience in Cooperation with Max Planck Society, 60528 Frankfurt am Main, Germany. ✉e-mail: sigurdsson@em.uni-frankfurt.de

patients and to investigate how such abnormalities relate to specific risk factors for the disease[32]. Although the symptoms of a disease as complex as schizophrenia are challenging to model in non-human animals—in particular hallucinations and delusions—sensory processing deficits can be readily examined in animal models using tests very similar to those used in patients[33,34]. Previously, we established an experimental paradigm, closely modeled on studies in human subjects and schizophrenia patients[6,19], for examining the processing of self-generated sounds in mice and found that responses to such sounds are attenuated in the mouse auditory cortex, similar to what is seen in the human brain[9] (see also refs. [10,11,35]). Studies have also begun to reveal the neural circuits that might underlie this attenuation in the auditory cortex, most notably projections from the motor cortex that could relay 'corollary discharge' signals to the auditory cortex[36,37]. Collectively, these studies provide a methodological and theoretical framework with which the processing of self-generated sounds could be investigated in animal disease models, thus yielding insights into the putative mechanisms underlying hallucinations and delusions.

In the current study, we examined the processing of self-generated sounds in the auditory cortex of $Df(16)A^{+/-}$ mice, which model one of the largest genetic risk factors for schizophrenia, the 22q11.2 microdeletion[38,39]. We found that whereas auditory cortical neurons in $Df(16)A^{+/-}$ mice respond normally to randomly-generated sounds, they show a diminished attenuation of responses to self-generated sounds. This deficit was most pronounced in lower cortical layers which we show are a major target of top-down projections from the motor cortex. Notably, these projections were decreased in $Df(16)A^{+/-}$ mice, suggesting an anatomical basis for the weaker attenuation of self-generated sounds. Consistent with this, motor-related signals in the auditory cortex of $Df(16)A^{+/-}$ mice were also reduced. Our results thus reveal how a major genetic risk factor for schizophrenia disrupts the filtering of self-generated sounds at the single-neuron level and suggest that this deficit results from altered corollary discharge signaling from motor to auditory cortex.

## Results

### Reduced attenuation of responses to self-generated sounds in $Df(16)A^{+/-}$ mice

In order to examine the processing of self-generated sounds in $Df(16)A^{+/-}$ mice we employed a paradigm we had previously used to examine processing of self-generated sounds in the mouse auditory cortex[9]. Briefly, animals were first trained to press a lever using operant conditioning. Once animals were lever pressing reliably, we coupled the lever with a sound generator so that each lever press triggered the delivery of a white noise auditory stimulus. The same sound was also presented randomly in the background, allowing us to compare neuronal responses to the same physical stimulus when it was randomly generated and self-generated (Fig. 1a; see Methods). $Df(16)A^{+/-}$ mice learned to lever-press at the same rate as their wild-type littermates and generated the same number of sounds during the recording sessions (Supplementary Fig. 1). While animals generated sounds by lever pressing we recorded the activity of neurons in the auditory cortex using multisite silicon probes spanning both upper and lower cortical layers (Fig. 1b; see Methods).

Consistent with previous findings[9-11,35], auditory cortex (ACx) neurons from wild-type mice displayed attenuated responses to self-generated sounds (Fig. 1c, e). Across all auditory-responsive neurons (187 of 488 from 9 mice; see Methods), responses to self-generated sounds were significantly weaker than to randomly-generated sounds (Fig. 1e, $p < 0.0001$, sign-rank test); furthermore, 47.59% or 89 of 187 of auditory responsive neurons displayed significantly weaker responses to self-generated sounds ($p < 0.05$, rank-sum test) whereas only three neurons displayed enhanced responses to these sounds. In $Df(16)A^{+/-}$ mice 38.70% (197 of 509 from 9 mice) of ACx neurons were auditory

responsive, a fraction comparable to that observed in wild-type animals ($p = 0.95$, Fisher's exact test). Attenuated responses to self-generated sounds were also observed in ACx neurons of $Df(16)A^{+/-}$ mice (Fig. 1d, f; $p < 0.0001$, $n = 197$, sign-rank test) but the attenuation appeared less pronounced than in wild-type mice. To compare the strength of attenuation across genotypes, we computed a modulation index (MI; Methods) whose values range from $-1$ (response only to the random sound) to 1 (response only to the self-generated sound) with 0 indicating equal responses to both stimuli. The MI values were significantly less negative in $Df(16)A^{+/-}$ mice than in their wild-type littermates (Fig. 1g; $Df(16)A^{+/-}$: $-0.08 \pm 0.02$, wild-type: $-0.25 \pm 0.02$; $p < 0.0001$, rank-sum test; $p = 0.0072$, hierarchical bootstrap; animal-averaged values $-0.09 \pm 0.04$ vs $-0.24 \pm 0.05$, $p = 0.014$, rank-sum test), indicating weaker attenuation of self-generated sounds (these MI values correspond, respectively, to -15% and 40% smaller responses to self-generated sounds relative to random sounds).

Since animals repeatedly experienced random and self-generated sounds, sensory adaptation might have influenced the responses to these sounds as well as the differences we observed between $Df(16)A^{+/-}$ and wild-type mice. In order to minimize the effects of sensory adaptation, in the analysis presented above and elsewhere we excluded sounds from analysis that occurred within a time window (1 s) after the previous sound (see Methods). However, shortening or lengthening this time window did not appreciably alter the difference in MI values between the two genotypes (Supplementary Fig. 2A). We also examined sensory adaptation in a subset of animals by recording responses to pairs of random sounds separated by different interstimulus intervals, similar to previous studies[40] (see Methods). This revealed a similar magnitude and time course of adaptation in the two genotypes (Supplementary Fig. 3). Another possible confounding factor is differences in overall behavioral state during the occurrence of random and self-generated sounds. In order to minimize this difference, random sounds were always excluded from analysis if they occurred during periods of behavioral inactivity (see "Methods"). We also recalculated the MI values after including only random sounds that occurred shortly before a lever press or when the lever was held down, during which the behavioral state of the animal should be similar as during self-generated sounds. Under both conditions, MI values remained less negative in $Df(16)A^{+/-}$ compared to wild-type mice (Supplementary Fig. 2B, C).

Because differences in MI values could reflect differences in responses to random sounds, self-generated sounds or both, we next directly compared responses to these two stimulus types between genotypes. This revealed that responses to self-generated sounds were stronger in the $Df(16)A^{+/-}$ mice (Fig. 1h; $p < 0.01$, rank-sum test), consistent with a decreased attenuation of these sounds. In contrast, responses to the randomly generated sounds did not differ between genotypes (Fig. 1h; $p = 0.46$), indicating that sensory responsiveness per se is not perturbed in these mice (as also demonstrated by the similar ratios of auditory responsive neurons in the two genotypes). Analysis of individual neurons furthermore revealed that, in comparison to wild-type mice, fewer neurons in $Df(16)A^{+/-}$ mice showed significantly attenuated responses (35.03% or 69 of 197, $p = 0.01$, Fisher's exact test) whereas more neurons showed enhanced responses to self-generated sounds (8.63% or 17 of 197, $p = 0.002$, Fisher's exact test). We also examined how the attenuation of self-generated sounds developed during the course of the session. To this end we computed the MI separately for early and late blocks of self-generated sounds (see Methods). MI values were significantly more negative in the later block (Fig. 1i, j; block X genotype ANOVA: main effect of block $p < 0.001$; wild-type mice early vs later block: $p < 0.0001$, $n = 144$ neurons; $Df(16)A^{+/-}$ mice early vs late block: $p < 0.01$, $n = 169$ neurons, rank-sum test), indicating that the attenuation of self-generated sounds increases with experience, as the association between lever-pressing

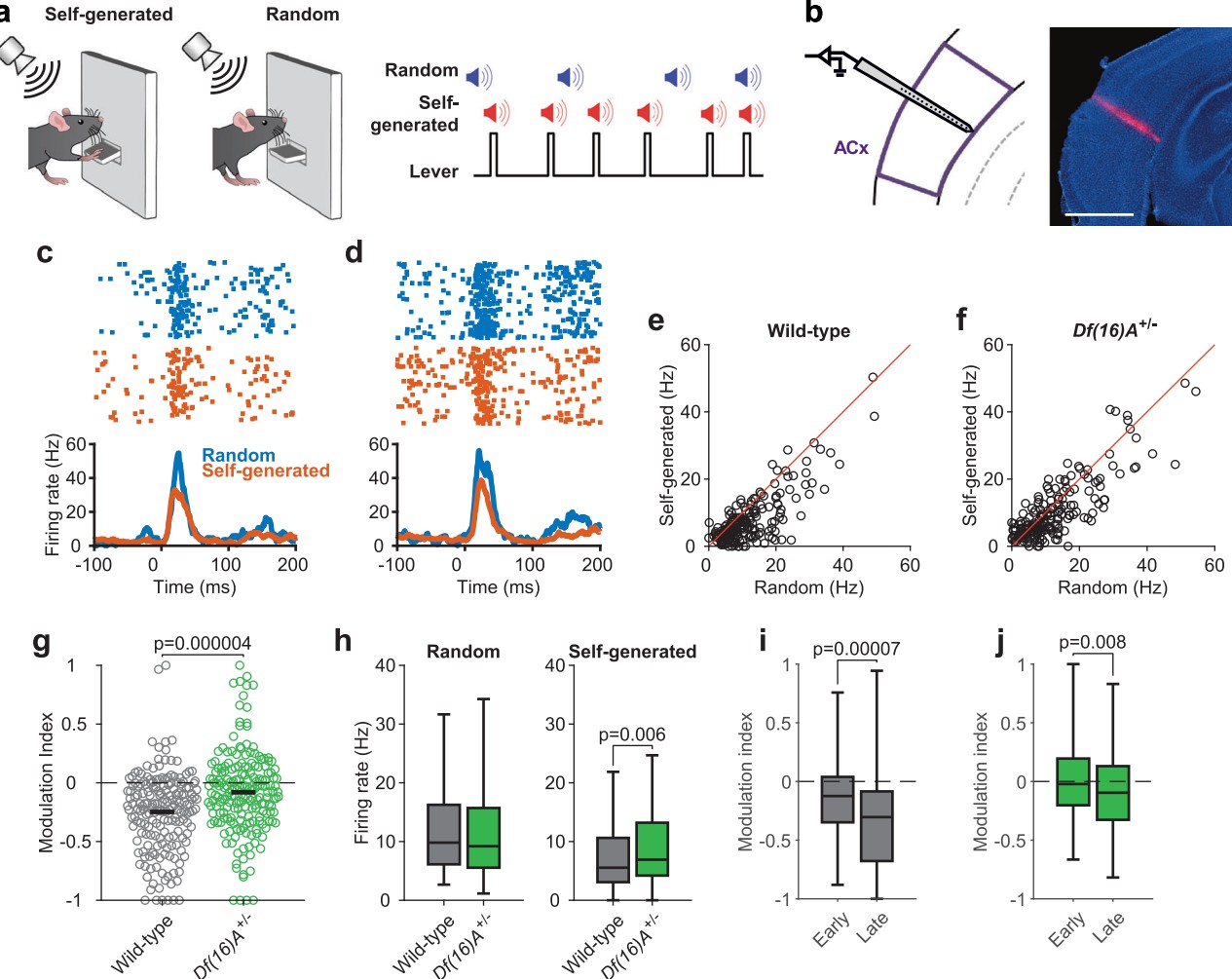

**Fig. 1 | Decreased attenuation of self-generated sounds in *Df(16)A*$^{+/-}$ mice. a** Mice pressed a lever which triggered the delivery of an auditory stimulus from an overhead speaker. The same sound was also delivered randomly with respect to the animals' behavior. Schematic reproduced from ref. 9. **b** recordings were made by inserting multi-site silicon probes into the auditory cortex (ACx), perpendicular to the cortical surface. Right, representative histological image showing a probe track labeled with a fluorescent dye. All animals used in this experiment (n = 18) had similar probe placement within ACx, with some variation along the anteroposterior axis. Scale bar: 1 mm. **c, d** Raster plots (top) and peri-stimulus time histograms (bottom) showing responses of auditory cortical neurons recorded from a wild-type (**c**) and a *Df(16)A*$^{+/-}$ (**d**) Mouse to random and self-generated sounds. **e, f** Responses to random and self-generated sounds in each neuron recorded from wild-type (**e**) and *Df(16)A*$^{+/-}$ (**f**) mice. Three neurons in (**e**) and one neuron in (**f**) are not visible due to the scale of the axes. **g** modulation index of each recorded neuron in the two genotypes. More negative values indicate stronger attenuation of the self-generated sound. Horizontal black lines indicate the mean in each genotype. **h** responses to random (left) and self-generated (right) sounds. Auditory cortical neurons from *Df(16)A*$^{+/-}$ mice showed a selective enhancement of responses to self-generated sounds. **i, j** responses to self-generated sounds in early and late blocks. Box plots represent the median (line), 25th and 75th percentiles (box), and 5th and 95th percentiles (whiskers). *P* values shown were calculated using a two-sided Wilcoxon rank-sum (**g, h**) or sign-rank (**i, j**) test. Data in **e–h** is from 197 neurons recorded from 9 *Df(16)A*$^{+/-}$ mice and 187 neurons recorded from nine wild-type mice; data in (**i, j**) is from 169 neurons recorded from 7 *Df(16)A*$^{+/-}$ and 144 neurons from 7 wild-type mice. Source data are provided as a Source Data file.

and sound-generation is learned. Furthermore, MI values were significantly less negative in *Df(16)A*$^{+/-}$ mice across blocks (main effect of genotype: $p < 0.0001$; no genotype X block interaction, $p = 0.37$). Overall, these findings demonstrate that *Df(16)A*$^{+/-}$ mice are impaired in attenuating responses to stimuli caused by their own behavior and reveal a possible cellular basis for similar deficits seen in schizophrenia patients using macroscopic measurements.

### Responses to self-generated sounds across cell types and cortical layers in *Df(16)A*$^{+/-}$ mice

Because inhibitory interneurons have been implicated in the modulation of auditory cortical neurons by movement[11,37], we next examined whether the deficit in attenuating self-generated sounds in *Df(16)A*$^{+/-}$ mice might be attributable to altered activity of inhibitory interneurons. To this end, we used a Gaussian mixture model to classify

recorded neurons as putative pyramidal neurons (pPNs) or interneurons (pINs) based on their spike waveform features (Methods; Fig. 2a). Of the neurons that could be classified with confidence ($p < 0.05$; 95.99% or 957/997 neurons), 14.0% were classified as putative inhibitory interneurons and the remaining 86.0% as putative pyramidal neurons. The ratio of pINs was similar in the two genotypes (Fig. 2b; wild-type: 12.88% of 466 cells; *Df(16)A*$^{+/-}$: 15.07% of 491 cells; $p = 0.35$, Fisher's exact test). The average firing rates of pINs were also comparable between genotypes (Fig. 2c; Wild-type: 7.64 ± 0.91 Hz, n = 60; *Df(16)A*$^{+/-}$: 5.88 ± 0.47 Hz, $n = 74$; $p = 0.54$, rank-sum test), as were the firing rates of pPNs (Fig. 2d; 4.41 ± 0.16 Hz, n = 406; *Df(16)A*$^{+/-}$: 4.69 ± 0.17 Hz, $n = 417$, $p = 0.39$, rank-sum test). It is possible that the weaker responses to self-generated sounds are caused by the recruitment of inhibitory interneurons and that this recruitment is impaired in *Df(16)A*$^{+/-}$ mice, resulting in enhanced responses to these sounds. If

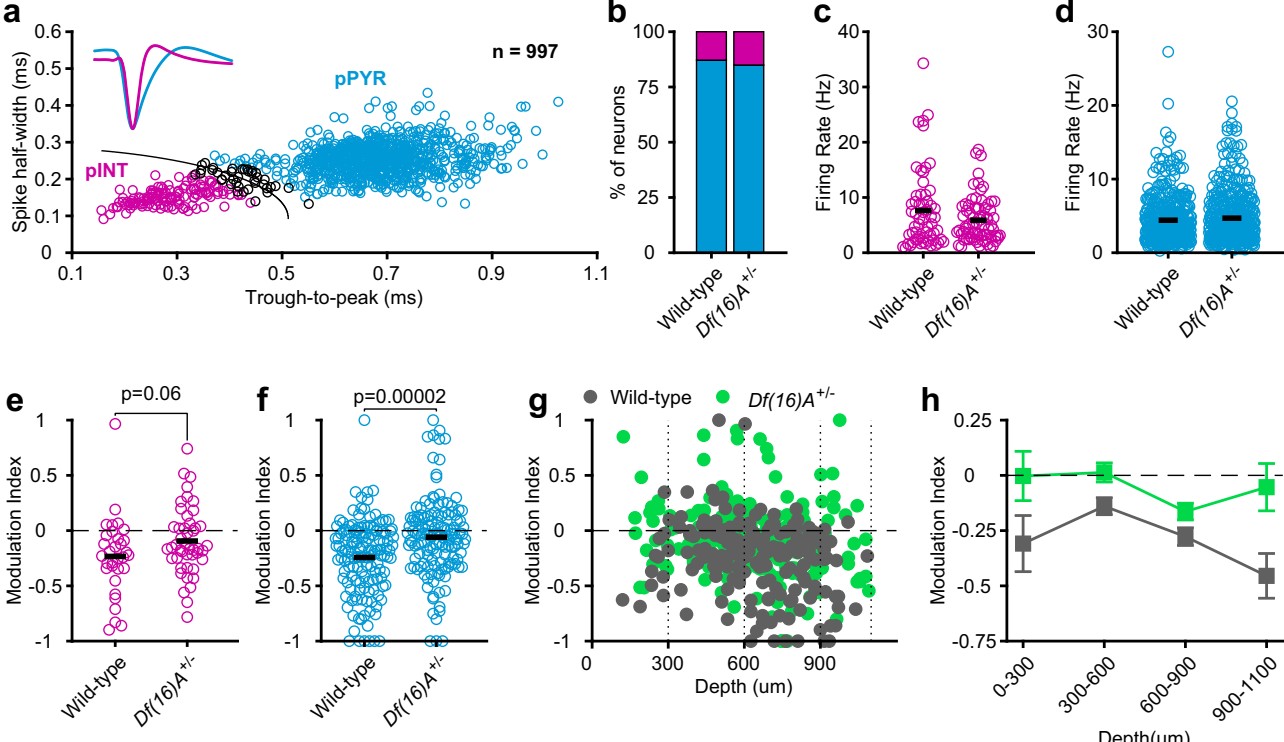

**Fig. 2 | Attenuation of self-generated sounds in *Df(16)A*^+/− mice across cell types and cortical layers. a** Trough-to-peak time and spike-half-width of all recorded neurons (combined from both genotypes). A 2-dimensional Gaussian mixture model was used to classify neurons as putative pyramidal (pPYR) and interneurons (pINT). Black line indicates classification boundary; black points indicate neurons with low classification confidence. Inset, mean ± s.e.m. waveforms of pPYRs and pINTs, normalized to the waveform trough. **b** ratios of pPYRs and pINTs in the two genotypes. **c, d** Average firing rates of pINTs (**c**; *Df(16)A*^+/−: *n* = 74, wild-type: *n* = 60) and pPYRs (**d**; *Df(16)A*^+/−: *n* = 417, wild-type: *n* = 406) in the two genotypes. **e, f** Modulation index of pINTs (**e**; *Df(16)A*^+/− *n* = 46, wild-type *n* = 32) and pPYRs (**f**; *Df(16)A*^+/−: *n* = 140, wild-type: *n* = 142). Horizontal black lines indicate the mean of each group. **g** modulation index of each neuron as a function of its depth from the cortical surface (*Df(16)A*^+/−: *n* = 197, wild-type: *n* = 187). **h**, Mean ± s.e.m. MI values of neurons in different depth bins (dotted lines in **g**). *P* values shown were calculated using a two-sided Wilcoxon rank-sum test. Data shown are from nine *Df(16)A*^+/− and nine wild-type mice, except in (**e**) from nine *Df(16)A*^+/− and eight wild-type mice. Source data are provided as a Source Data file.

this were the case, interneurons and pyramidal neurons would be expected to respond differently to self-generated sounds and furthermore, responses of interneurons should be altered in *Df(16)A*^+/− mice. To address this, we examined responses to self-generated sounds separately for pPNs and pINs in the two genotypes. Contrary to expectations, however, in both genotypes responses of pINs to self-generated sounds were attenuated to a similar degree as in pPNs (Fig. 2e, f; cell type X genotype ANOVA; no effect of cell type: *p* = 0.77; main effect of genotype: *p* = 0.00028; no cell type X genotype interaction: *p* = 0.77).

We next examined responses to self-generated sounds at different cortical depths (Fig. 2g). In wild-type mice, the strength of attenuation varied with depth from the cortical surface (*p* < 0.0008; One-way ANOVA) and was most pronounced at the deepest recording sites (Fig. 2h). The MI values of neurons recorded from *Df(16)A*^+/− mice, by contrast, appeared to show less variability as a function of cortical depth and were overall less negative than in wild-types at all depth bins (Fig. 2h). Accordingly, a depth by genotype ANOVA revealed a main effect of depth (*p* = 0.0001), genotype (*p* < 0.0001) as well as a depth X genotype interaction (*p* < 0.05). Notably, the deficit in *Df(16)A*^+/− mice was greatest at the deepest sites where attenuation was largest in wild-type mice (900–1100 μm; wild-type mice: −0.45 ± 0.08, *Df(16)A*^+/− mice: −0.05 ± 0.09, *p* < 0.01, rank-sum test). In fact, at this depth, modulation indices did not differ from zero in the *Df(16)A*^+/− mice (*p* = 0.45, sign-rank test) suggesting that the altered attenuation of self-generated sounds in *Df(16)A*^+/− mice is particularly pronounced in the lower cortical layers.

## Diminished motor preparatory activity in the auditory cortex of *Df(16)A*^+/− mice

What might be the physiological mechanism responsible for the weaker attenuation of self-generated sounds observed in *Df(16)A*^+/− mice? Considerable evidence suggests that processing of self-generated stimuli is influenced by 'corollary discharge signals', which are copies of motor commands that are relayed from motor areas to sensory areas of the brain[1,12,13]. We reasoned that such signals might manifest themselves in the activity of auditory cortical neurons prior to sound generation[37,41] and be disrupted in *Df(16)A*^+/− mice. Consistent with this, we observed that many neurons in the auditory cortex of wild-type mice displayed gradual changes in their activity during the time period leading up to the lever press (Fig. 3a). These changes consisted largely of increases in firing rate, as reflected in the population average (Fig. 3b). Auditory cortical neurons in *Df(16)A*^+/− mice also increased their firing rate prior to the lever press but to a lesser extent than in wild-type mice. Whereas firing rates were initially comparable between the genotypes (−1000 to −900 ms: Wild-type: 4.48 ± 0.36 Hz, *Df(16)A*^+/−: 4.28 ± 0.28 Hz, *p* = 0.92, rank-sum test), they began to diverge ~600 ms before the lever press (Fig. 3b). Confirming this, a genotype X time ANOVA revealed a main effect of genotype (*p* < 0.0001), time (*p* < 0.0001), as well as a time × genotype interaction (*p* < 0.0001). These genotype differences persisted until immediately before (0–200 ms) the lever press (Fig. 3c; rank-sum test, *p* < 0.0001; hierarchical bootstrap, *p* = 0.0012; animal-averaged values: 2.91 ± 0.57 vs 1.37 ± 0.21 Hz, *p* = 0.0078, rank-sum test). Overall, these results

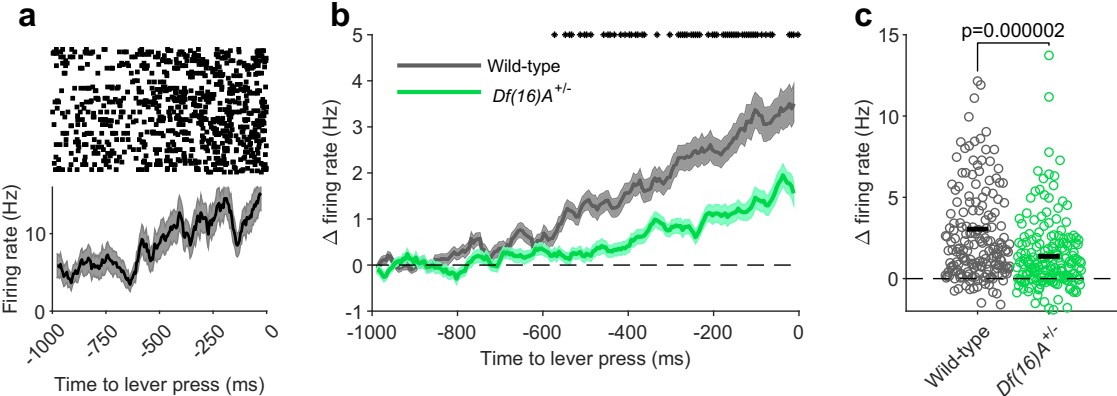

**Fig. 3 | Diminished motor preparatory activity in the auditory cortex of *Df(16)A*[+/−] mice. a** Raster plot (top) and peri-stimulus time histogram (bottom) showing the firing rate of an example neuron preceding a lever press. **b** Changes in firing rates during the time period preceding lever press (relative to average firing rates 900–1000 ms prior to lever press) in the two genotypes. Asterisks indicate time-points where firing rates differed significantly between genotypes (*p* < 0.01, two-sided Wilcoxon rank-sum test). **c** Average change in firing rates 0–200 ms before lever press in each neuron from the two genotypes. Ten neurons are not visible due to the scale of the y-axis. Horizontal black lines indicate the mean of each genotype. *P* value was calculated using a two-sided Wilcoxon rank-sum test. Shaded areas indicate mean ± s.e.m. across trials (**a**) or neurons (**b**). Data in (**b**–**d**) represent 183 neurons from 9 *Df(16)A*[+/−] mice and 189 neurons from 9 wild-type mice. Source data are provided as a Source Data file.

suggest that the signaling of motor-related information to the auditory cortex is decreased in *Df(16)A*[+/−] mice.

## Top-down projections from motor cortex modulate responses in auditory cortex

In the mammalian auditory system, one source of motor-related inputs that could provide corollary discharge signals for the processing of self-generated sounds are direct projections from the secondary motor cortex to the auditory cortex[36,42–44]. To confirm this, we injected a virus expressing green fluorescent protein (GFP) in the secondary motor cortex (M2; Fig. 4a). This labeled M2 neurons' axon terminals throughout the brain, including in the auditory cortex (Fig. 4b, c). Interestingly, GFP-positive axon terminals were not uniformly distributed across layers of the auditory cortex but were particularly concentrated in the deep cortical layers as well as superficial layers (Fig. 4c, d). The robust projection to the deep layers is particularly interesting given that this is where we observed the strongest attenuation of self-generated sounds (Fig. 2h; see also ref. 9).

We next investigated how motor cortical projections influence the activity of auditory cortical neurons. To this end, we expressed channelrhodopsin-2 in M2 neurons and stimulated their axon terminals in the auditory cortex with light while simultaneously recording the activity of auditory cortical neurons (Fig. 4e). Of all recorded neurons (*n* = 886 from 22 recordings in 4 mice), ~4% (38/886) responded to light delivery with short latency (<10 ms; see Methods) and temporally high precision to terminal stimulation, suggesting that they receive monosynaptic input from M2 (Fig. 4f, g and Supplementary Fig. 4A, E). Strikingly, these directly activated neurons were exclusively located in the lower cortical layers (>600 µm; Fig. 4i), consistent with the higher density of axonal projections in these layers (Fig. 4d). In many of these neurons, the initial excitation was followed by inhibition (e.g., Fig. 4g and Supplementary Fig. 4E), which was sometimes also seen in the absence of short-latency excitatory responses (Fig. 4h). Neurons displaying inhibitory responses (see Methods) were also more widely distributed across layers (Fig. 4i), comprising 4.64% (9/194) of upper layer neurons (<600 µm below brain surface) and 7.80% (54/692) of lower layer neurons (>600 µm below brain surface; *p* = 0.16, Fisher's exact test; see also Supplementary Fig. 4A). Similar results were obtained when stimulating M2 cell bodies in a subset of recording sessions (*n* = 220 neurons; Supplementary Fig. 4B–D). Interestingly, neurons in deep layers of the auditory cortex can inhibit activity in other layers[45], suggesting a mechanism whereby the selective activation of deep layer neurons by M2 inputs could lead to more widespread inhibition across cortical layers.

Light can affect neural activity independently of ChR2 expression, for example through increases in brain temperature[46,47] or by eliciting visual responses, which can be observed in the lower layers of the auditory cortex[48]. Light can also cause photoelectric artifacts that may interfere with the detection of action potentials. To examine whether these confounding variables could have influenced our results, we delivered light in the same manner to the auditory cortex of mice that did not express ChR2 in M2. This did not cause any visible changes in the firing rates of auditory cortex neurons (*n* = 161 from 12 recordings in 3 mice; Supplementary Fig. 4F). Notably, none of the recorded neurons displayed short-latency excitatory responses to light and only 1 neuron showed inhibitory responses, using the same response criteria as for ChR2-expressing mice (Methods). Thus, the responses we observed in mice expressing ChR2 were caused by the optogenetic activation of M2 axon terminals in the auditory cortex.

## The auditory cortex of *Df(16)A*[+/−] mice receives fewer top-down projections from motor cortex

Based on the above findings, we hypothesized that projections from the motor cortex to the auditory cortex might be disrupted in *Df(16)A*[+/−] mice. To test this, we injected a retrograde virus expressing GFP into the auditory cortex of *Df(16)A*[+/−] mice and their wild-type littermates (Fig. 5a, b). We then quantified the number of retrogradely labeled neurons in frontal areas using a semi-automatic analysis pipeline for cell detection and alignment of brain sections to a reference atlas (see Methods). Labeled neurons were found in many frontal regions including motor cortical areas (primary and secondary), the medial prefrontal cortex (cingulate, prelimbic and infralimbic subregions), somatosensory cortex, the insula and claustrum (Fig. 5c–g), in agreement with previous studies[36,42–44]. We first compared the strength of inputs from these areas to the auditory cortex by quantifying the number of retrogradely labeled neurons in wild-type mice (*n* = 9). The largest number of neurons was found in the secondary motor cortex and the orbitofrontal cortex, followed by the primary motor cortex, insula and other areas (Fig. 5h). Notably, 31.23 ± 3.15% of all retrogradely labeled neurons were located in the motor cortical regions combined, making them a major source of top-down inputs to the

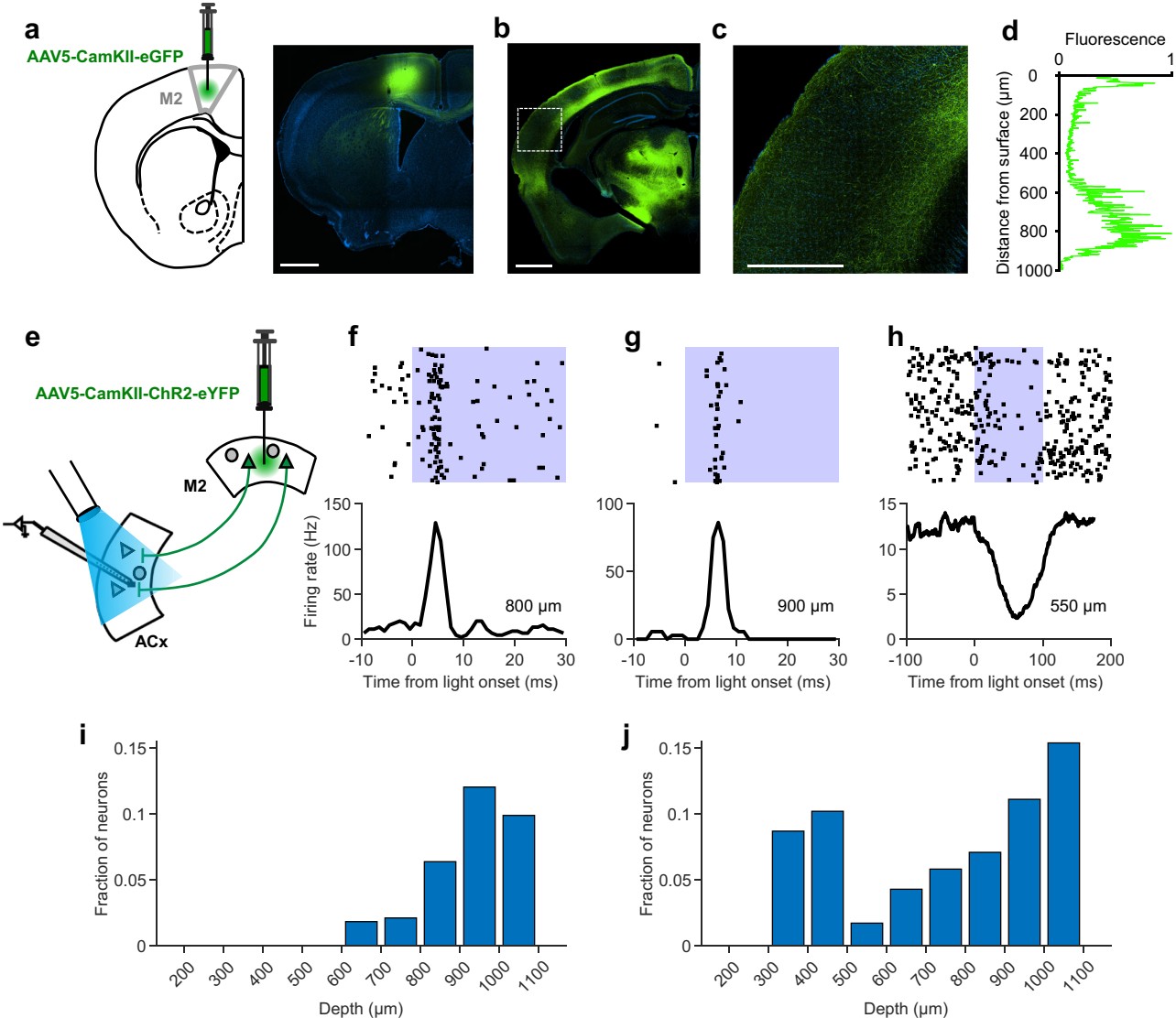

**Fig. 4 | Top-down projections from the motor cortex influence activity in the auditory cortex. a** A virus expressing GFP was injected into the secondary motor cortex (M2; gray outlined area). Histological image on the right shows the expression of GFP in M2. **b** GFP-expressing terminals of M2 neurons at the anteroposterior level of the auditory cortex (dash-outlined area). **c** Higher-magnification view of the auditory cortex (dash-outlined area in **b**). Scale bar represents 1 mm in (**a**, **b**) and 0.5 mm in (**c**). **d** GFP expression as a function of depth from the cortical surface. Note the strong expression in lower cortical layers. **e** A virus expressing Channelrhodopsin-2 (ChR2) was injected into M2 and the terminals of M2 neurons were optogenetically stimulated while at the same time recording the activity of auditory cortical neurons. **f**–**h** Raster plots and peri-stimulus time histograms of three auditory cortical neurons responding to optogenetic stimulation of M2 axon terminals. Blue shaded area denotes the time period of light delivery. Note the short-latency excitatory responses in (**f**) and (**g**), indicating monosynaptic input from M2. Numbers on the bottom right of the PSTHs indicate the depth of the neurons below the brain surface. **i** Fraction of neurons showing short-latency excitatory responses (as exemplified by the neurons in (**f**) and (**g**)) as a function of their depth below cortical surface. **j** Fraction of neurons showing inhibitory responses to M2 terminal stimulation (as exemplified by the neuron in (**h**)). Fractions in (**i**) and (**j**) were calculated relative to the total number of neurons at each depth. Data in (**a**–**d**) is from one mouse; similar results were obtained from five additional mice injected with a virus expressing GFP or ChR2 in M2. Data in (**i**, **j**) are from 886 neurons recorded from four wild-type mice. Source data are provided as a Source Data file.

auditory cortex. Projections were stronger from the hemisphere ipsilateral to the injection site and originated mostly in the lower cortical layers (Fig. 5i).

We next compared the strength of frontal inputs to the auditory cortex between *Df(16)A*^+/− mice (*n* = 10) and their wild-type littermates (*n* = 9). Overall, fewer retrogradely labeled neurons were observed in frontal brain regions of *Df(16)A*^+/− mice (main effect of genotype in a area × genotype ANOVA: *p* < 0.0001) but this effect varied between regions (area × genotype interaction: *p* < 0.001). Consistent with our hypothesis, in *Df(16)A*^+/− mice we observed fewer labeled neurons in both the primary (p = 0.01, rank-sum test) and secondary (*p* = 0.003) motor cortices compared to wild-type littermates, suggesting that

motor cortical projections to the auditory cortex are decreased in these mice (Fig. 6a). These genotype differences were observed in both the ipsilateral and contralateral hemispheres (Supplementary Fig. 5A, B). We also observed fewer labeled neurons in the anterior cingulate cortex (*p* = 0.01), specifically in its dorsal subdivision (Supplementary Fig. 5C). In the orbitofrontal cortex overall, differences in cell counts did not quite reach statistical significance (*p* = 0.0535), although fewer labeled neurons were observed in specific subdivisions (Supplementary Fig. 5C). No genotype differences were observed for other frontal cortical regions (Fig. 6a; all *p* > 0.17). Furthermore, we observed comparable numbers of retrogradely labeled neurons in auditory cortical areas contralateral to the injection site (Fig. 6b, c; all *p* > 0.4). This

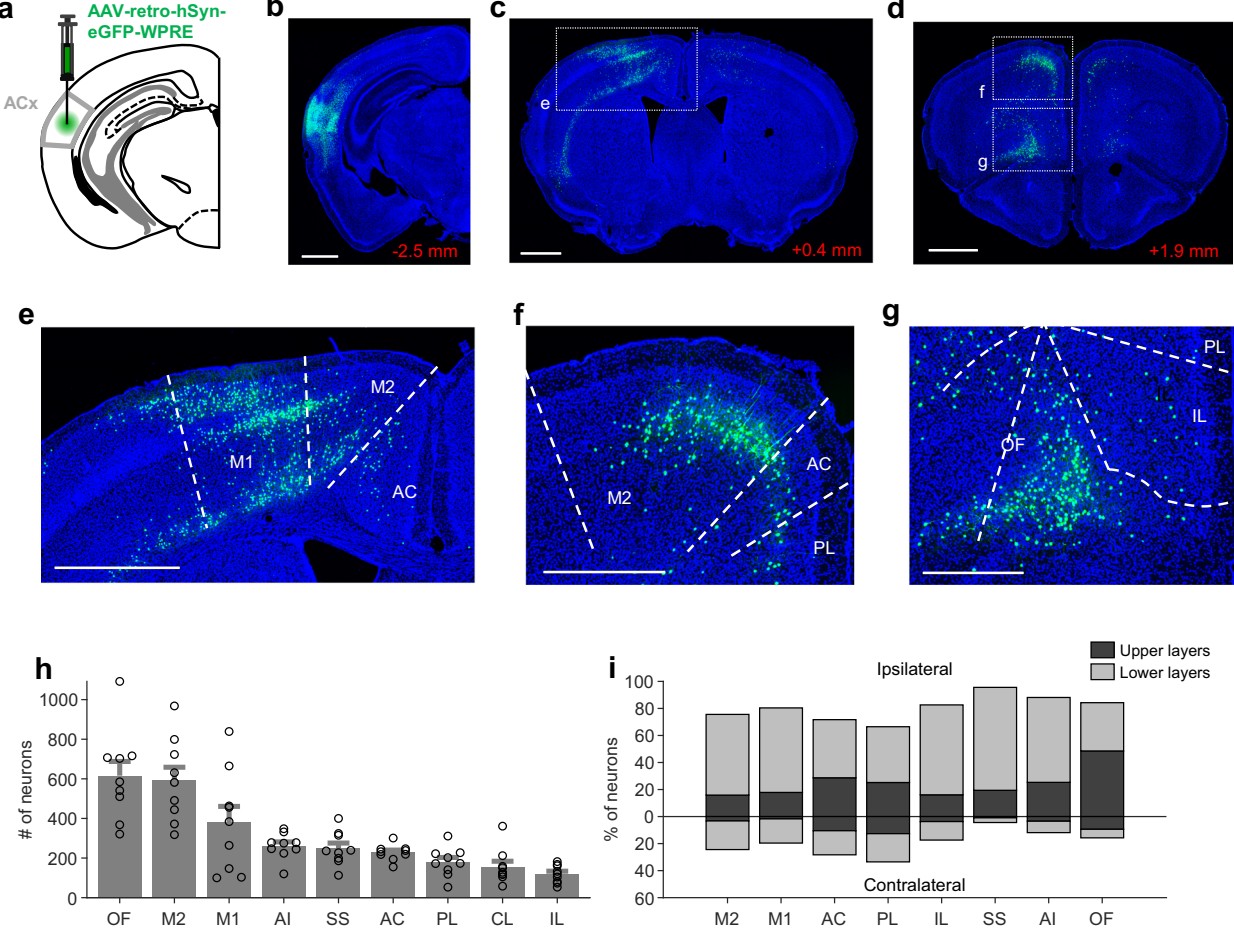

**Fig. 5 | Quantification of frontal cortical inputs to the auditory cortex. a** A retrogradely-traveling virus expressing GFP was injected into the auditory cortex. **b–d** Histological example showing retrogradely labeled cells at the site of injection (**b**) and in the frontal cortex at two anteroposterior levels (**c**, **d**). Numbers in red indicate position anterior to bregma. **e-g** Areas enclosed by rectangles in (**c**, **d**) are shown at a higher magnification. Scale bars represent 1 mm in (**b–e**) and 0.5 mm in (**f**, **g**). Area boundaries correspond to the Allen Mouse Brain Atlas. **h** Number of retrogradely labeled neurons in different frontal regions in wild-type mice. Error bars represent mean ± s.e.m. across mice. **i** Distribution of cells in each frontal cortical region across hemispheres (relative to injected hemisphere) and cortical layers (upper: layers 1-4; lower: layers 5–6). Percentages are relative to the total number of retrogradely labeled neurons in each brain region. M2, secondary motor cortex; M1, primary motor cortex; AC, anterior cingulate cortex; PL, prelimbic cortex; IL, infralimbic cortex; CL, claustrum; SS, somatosensory cortex; AI, anterior insula; OF, orbitofrontal cortex. Data in (**h**, **i**) is from nine wild-type mice. Source data are provided as a Source Data file.

demonstrates that the fewer projections from the motor cortices do not simply reflect a general nonspecific disruption in long-range inputs to the auditory cortex. Finally, we found that the number of retrogradely labeled neurons in the medial geniculate nucleus of the thalamus, the main source of bottom-up sensory input to the auditory cortex, was also comparable between *Df(16)A*[+/−] mice and their wild-type littermates (Fig. 6d, e, *p* = 1.00). This is consistent with our finding that responses to randomly generated sounds were not disrupted in the *Df(16)A*[+/−] mice (Fig. 1h). Taken together, these results demonstrate that the auditory cortex of *Df(16)A*[+/−] mice receives decreased top-down input from motor areas of the cortex. Since these inputs supply corollary discharge signals to the auditory cortex, their impairment could underlie the deficit in attenuating responses to self-generated sounds observed in these mice.

## Discussion

Deficits in anticipating and attenuating the sensory consequences of behavior have consistently been observed in schizophrenia patients. In the auditory domain, patients fail to attenuate responses to their own speech[3,14–16] or manually triggered sounds[17–19] and similar deficits have been revealed in the somatosensory[20,49] and visual[21,50] modalities. These deficits could underlie the hallucinations and delusions that are

characteristic of the disease, but whose causes are poorly understood. Hallucinations, which typically take the form of hearing voices, could emerge if inner speech is not recognized as self-generated but is rather misattributed to an external source[24–26,51]. Delusions of control, whereby patients experience their actions as caused by outside forces, could also result from a failure to correctly recognize the sensory consequences of behavior[49,51]. Indeed, several studies have reported a correlation between deficits in predicting the sensory consequences of actions and the severity of hallucinations and delusions in schizophrenia patients[27–31]. Such sensory prediction deficits could also underlie the failures of self-recognition that are observed in schizophrenia patients[52].

Whereas many of the sensory deficits seen in schizophrenia have been examined in animal models, to date such models have not been used to investigate the impairments in attenuating self-generated stimuli. If such impairments underlie hallucinations and delusions, studying them in animal models may yield insights into the neural mechanisms of these symptoms, which are otherwise difficult if not impossible to model in non-human animals (but see ref. 53). Importantly, animal models can help reveal how sensory impairments manifest at the cellular level and how they relate to the risk factors of the disease. To this end, in the current study we examined the processing

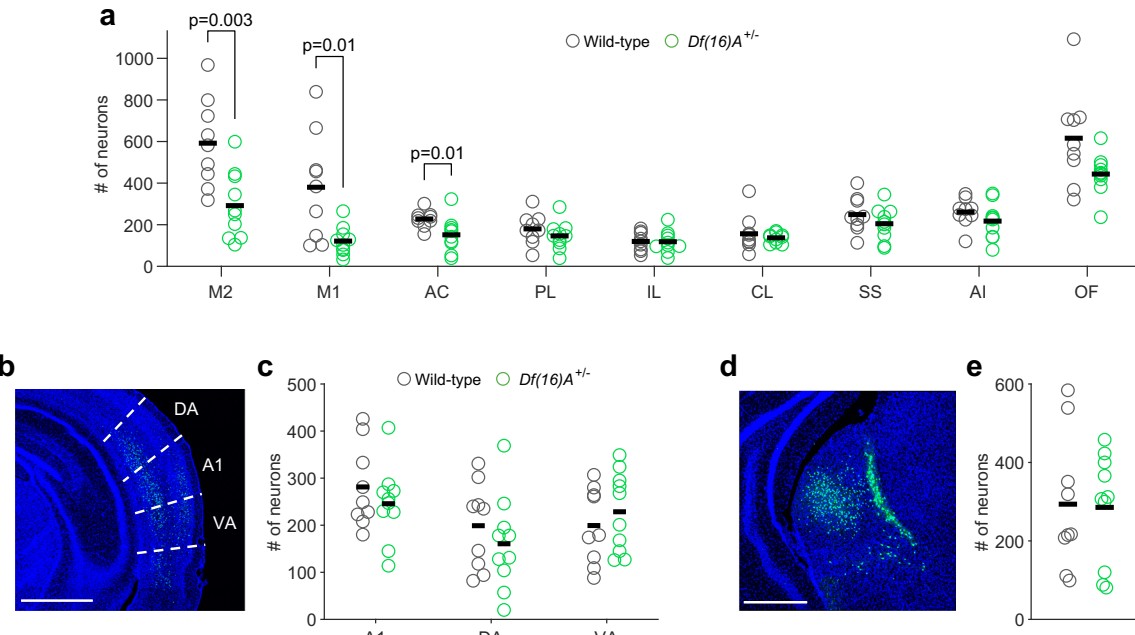

**Fig. 6 | Decreased motor cortical inputs to the auditory cortex in *Df(16)A*⁺/⁻ mice. a** Number of retrogradely labeled neurons in each frontal region in *Df(16)A*⁺/⁻ and wild-type mice. **b** Histological image showing retrogradely labeled neurons in auditory cortical regions contralateral to the injection site. **c** Number of labeled neurons in contralateral auditory cortical regions in the two genotypes. Area boundaries correspond to the Allen Mouse Brain Atlas. **d** Retrogradely labeled neurons in the auditory thalamus. **e** Number of retrogradely labeled neurons in the auditory thalamus in the two genotypes. Scale bars in (**b**) and (**d**) represent 1 and

0.5 mm, respectively. Horizontal black lines indicate the average number of neurons across mice in each genotype. M2, secondary motor cortex; M1, primary motor cortex; AC, anterior cingulate cortex; PL, prelimbic cortex; IL, infralimbic cortex; CL, claustrum; SS, somatosensory cortex; AI, anterior insula; OF, orbitofrontal cortex; A1, primary auditory cortex; DA, dorsal auditory cortex; VA, ventral auditory cortex. Data in (**a**), (**c**), and (**e**) is from 10 *Df(16)A*⁺/⁻ and nine wild-type mice. P values shown were calculated using a two-sided Wilcoxon rank-sum test. Source data are provided as a Source Data file.

of self-generated sounds in the auditory cortex of *Df(16)A*⁺/⁻ mice[38], which model one of the largest genetic risk factors for schizophrenia, the 22q11.2 microdeletion[39]. Consistent with previous findings[9–11,35] we found that neurons in the auditory cortex of wild-type mice showed attenuated responses to sounds generated by their own behavior. In *Df(16)A*⁺/⁻ mice, however, this attenuation was weaker in magnitude, recapitulating at the cellular level findings obtained in schizophrenia patients using macroscopic measurements of brain activity[14,18,19]. It remains to be determined whether this electrophysiological alteration is associated with perceptual or behavioral deficits in the *Df(16)A*⁺/⁻ mice. Behavioral experiments measuring detection thresholds of random and self-generated sounds in *Df(16)A*⁺/⁻ mice, or their ability to discriminate these sounds behaviorally, could help address this question.

Although deficits in the processing of self-generated stimuli have not been examined in schizophrenia patients carrying the 22q11.2 microdeletion specifically, it is worth pointing out that the clinical symptoms and structural brain abnormalities seen in these patients are largely similar to those of schizophrenia patients in general[54]. Patients carrying the 22q11.2 microdeletion would therefore also be expected to display deficits in processing self-generated stimuli, but this remains to be tested. Human 22q11.2 carriers also display mild conductive hearing loss[55], but this has not been consistently replicated in 22q11.2 mouse models[56–58]. These discrepancies are most likely due to differences in laboratory conditions since hearing loss, when present in 22q11.2 mouse models, is caused by increased susceptibility to middle ear infection[59]. Although we did not test for hearing loss in the *Df(16)A*⁺/⁻ mice, it is worth noting that hearing loss has been shown to alter the balance of excitation and inhibition in the auditory cortex as well as the firing rates of cortical neurons[60]. However, we found that the average firing rates of auditory cortical neurons, as well as their responses to random sounds, were not altered in *Df(16)A*⁺/⁻ mice. Furthermore, these

mice did not differ from their wild-type littermates in terms of sensory adaptation to auditory stimuli or the number of thalamic neurons projecting to the auditory cortex. Taken together, these results argue against the possibility that the deficits in processing self-generated sounds in *Df(16)A*⁺/⁻ mice arise from a more generalized auditory dysfunction.

What mechanism might underlie the reduced attenuation of self-generated sounds in *Df(16)A*⁺/⁻ mice? To gain insight into this we examined the activity of putative inhibitory interneurons, which were identified based on their narrow spike waveform. We found that the ratios of putative interneurons as well as their firing rates were comparable in both genotypes. Furthermore, the deficit in attenuating responses to self-generated sounds was similar for both putative interneurons and pyramidal neurons from *Df(16)A*⁺/⁻ mice. However, given the uncertainties involved in identifying interneurons from extracellular waveform features and the fact that our waveform criteria apply to one specific interneuron subtype (most likely parvalbumin-expressing)[61], these results still leave open the possibility of interneuron dysfunction in the auditory cortex of *Df(16)A*⁺/⁻ mice. Indeed, interneuron dysfunction has been consistently observed both in patients as well as animal models of the disease[32,62] and interneurons also likely play a role in the attenuation of self-generated sounds[11].

Alternatively, the weaker attenuation in *Df(16)A*⁺/⁻ mice could reflect disrupted top-down inputs to the auditory cortex. Notably, we found that auditory cortical neurons increased their activity in the time period preceding the self-generation of sounds, consistent with previous studies that have observed activity changes in ACx preceding movement onset[37,41]. However, this preparatory motor activity was reduced in *Df(16)A*⁺/⁻ mice. Such preparatory motor activity might reflect 'corollary discharge' signals, which are copies of motor commands that are conveyed to sensory areas and which can modulate responses to the self-generated sensory input[1,12,13,63,64]. Such signals are

likely conveyed to the auditory cortex by monosynaptic projections from M2[36,42–44]. Indeed, we found that M2, together with the orbito-frontal cortex[65] constituted the largest source of inputs to the auditory cortex from frontal areas. In agreement with previous studies, M2 axonal projections were particularly enriched in the lower layers (likely 5 and 6) of the auditory cortex. Consistent with this, ACx neurons that responded at a short latency to optogenetic stimulation of M2 axon terminals (and are thus likely recipients of monosynaptic input from M2), were found exclusively in the lower cortical layers. This finding is notable since it is in the lower layers where we observed the strongest attenuation of responses to self-generated sounds in wild-type mice and the largest deficit in Df(16)A[+/−] mice. We also observed inhibitory responses to M2 terminal stimulation which were more widely distributed throughout the layers of auditory cortex, possibly mediated by translaminar connections originating in deeper layers[45].

Based on the above findings, we hypothesized that the deficit in attenuating responses to self-generated sounds in Df(16)A[+/−] mice might be due to weaker inputs from M2 to the auditory cortex. To this end, we examined projections to the ACx in these mice from all frontal cortical regions. Consistent with our hypothesis, we observed fewer projections from M2 as well as the primary motor cortices in Df(16)A[+/−] mice. Fewer inputs were also observed from the cingulate cortex, which is known to exert top-down control over processing in sensory areas[44]. However, inputs from many other frontal regions, such as the prelimbic cortex and the insula, were not affected, arguing against a generalized deficit in the top-down control of the auditory cortex. Likewise, the strength of inputs from the auditory thalamus was comparable between the two genotypes, consistent with normal responses in auditory cortical neurons to randomly-generated sounds. These results complement and extend previous findings of long-range connectivity deficits in 22q11.2 mouse models that have been revealed using measures of anatomical[66,67] as well as functional[67,68] connectivity. Although the molecular mechanisms underlying the reduced M2-ACx connectivity in Df(16)A[+/−] mice remain to be determined, possible candidates include impaired axonal growth[67], excessive synaptic pruning[69] and oxidative stress[66]. Abnormal connectivity has also been consistently observed in schizophrenia patients and is thought to be a central pathophysiological mechanism of the disease[70]. Notably, structural abnormalities have been observed in pathways that could provide corollary discharge signals during speech production[71], and these abnormalities correlate with deficits in attenuating responses to speech sounds in patients[72]. Our results demonstrate how such corollary discharge pathways can be disrupted by a specific genetic risk factor for schizophrenia, thus impairing the ability to distinguish self-generated from externally generated sensory input.

## Methods

### Animals
Nineteen male Df(16)A[+/−] mice[38] bred on a C57BL/6N background and 18 of their male wild-type littermates were used in the study. Nine Df(16)A[+/−] mice and nine wild-type littermates were used for the electrophysiological experiments described in Figs. 1–3; 10 Df(16)A[+/−] mice and nine wild-type littermates were used for the anatomical tracing experiments shown in Figs. 5, 6. In addition, nine male C57/Bl6N mice (Charles River Laboratories) were used for the anatomical and optogenetic experiments shown in Fig. 4 and Supplementary Fig. 4. Due to the time-intensive nature of the experiments and limited resources, examining sex differences was beyond the scope of the study. Animals were 7–16 weeks old at the beginning of the experiments and were housed in individual cages inside a ventilated animal cabinet (Scantainer, Scanbur; ambient temperature: 20–24 °C; humidity: 40–65%). Animals were maintained on a 12-h light/dark cycle (lights on at 8 a.m.) and all experiments were performed during the light phase. All procedures were approved by the local animal care committee (TVA FU-1256, Regierungspräsidium Darmstadt, Germany).

### Surgical procedures
Mice were anesthetized in a chamber filled with 3% isoflurane and placed in a stereotaxic frame. Prior to the beginning of surgery, animals were injected with carprofen (4 mg/kg, subcutaneously) and dexamethasone (2 mg/kg, subcutaneously) for reducing pain and inflammation; atropine (50 μl, intraperitoneal) to decrease mucus secretions; and Ringer's solution (0.8 ml, subcutaneously) as fluid replacement. Lidocaine gel (2% lidocaine hydrochloride; Emla, Astra-Zeneca) was applied on the scalp as a local anesthetic. An incision was then made in the scalp to expose the skull and remove the overlying connective tissue.

In animals in which recordings were to be performed from the auditory cortex, a stainless-steel headpost (Luigs and Neumann, Ratingen, Germany) was cemented to the skull behind lambda. Small screws (MF-5182, BaSi) with DIP socket pins attached (#AR 40-HZL-TT, Assmann WSW Components) were inserted into the skull overlying the frontal cortex and cerebellum to serve as reference and ground, respectively. An additional screw was inserted in the skull over the right frontal cortex to provide additional anchoring support. The skull was covered with cement except for the area overlying the auditory cortex which was marked with a waterproof pen (for a craniotomy performed later) and covered with a biocompatible silicone elastomer (Kwik-Sil, World Precision Instruments).

In animals in which inputs to the auditory cortex were examined, a craniotomy was made above the left auditory cortex (2.65 mm posterior to bregma, 4.25 mm lateral to bregma). A syringe (NanoFil, 10 μl, World Precision Instruments) attached to a blunt 35 gauge needle containing a retrogradely-traveling virus expressing enhanced green fluorescent protein (AAV-retro-hSyn-eGFP-WPRE, Vector Biolabs, $10 \times 10^{13}$ vg/ml) was then inserted through the craniotomy to a depth of 1.25 mm below skull surface at bregma. A total of 150 nl of viral construct was then infused at a constant speed of 33 nl/min, controlled by a micro-syringe pump (UltraMicroPump, World Precision Instruments) and a pump controller (Micro4, World Precision Instruments). In order to ensure diffusion of the viral construct into the tissue, the needle was left in place for 10 min after the end of the infusion, then withdrawn by 0.05 mm and left in place for another 5 min, before being removed from the brain. The scalp was then sutured closed and the animals allowed to recover from the surgery. Four weeks following the virus injection, animals were euthanized, their brains removed and histologically processed (see section 'Histology').

In animals in which inputs from secondary motor cortex (M2) were examined or optogenetically stimulated a craniotomy was made over M2 (1.3 mm anterior to bregma and 0.8 mm lateral to midline). A syringe (NanoFil, 10 mL, World Precision Instruments) attached to a 35 gauge needle was then inserted into M2. A virus expressing green fluorescent protein (GFP; AAV5-CaMKIIa-GFP, University of North Carolina Vector core) was injected in two mice and a virus expressing Channelrhodopsin-2 (ChR2; AAV5-CamKIIa-hChR2(H134R)-EYFP, University of North Carolina Vector Core) was injected in four mice. In each animal, two injections of 150 nl each were made at 0.35 and 0.75 mm below the brain surface at a rate of 30 nl/min. Injections were performed using a micro-syringe pump (UltraMicroPump, World Precision Instruments) and a pump controller (Micro4, World Precision Instruments). Syringes were left in place for 10 min before being withdrawn. The scalp was then sutured closed and lidocaine was applied on the stitches to reduce potential post-surgical pain. Five to six weeks after virus injection, GFP-injected mice were sacrificed and histologically examined (see "Histology", below). ChR2-injected mice underwent another surgery 5–7 weeks after virus injection in order to prepare them for head-fixed recordings, as described above.

During all surgeries, anesthesia was maintained with an isoflurane concentration of 1–2% (in oxygen at a flow rate of 0.35 l/min), which was regularly adjusted based on the monitored breathing rate (i.e., isoflurane concentration was decreased if breathing rate fell below

1 Hz). Body temperature was maintained at 37 °C with a heating blanket placed under the animal. Animals were allowed to recover for at least 1 week following surgery.

### Recording of neural activity during self-generation of auditory stimuli in head-fixed mice

Following recovery from surgery, animals were handled and their water intake was restricted to 1 mL per day, which resulted in their weight decreasing to ~80–85% of their original weight over a period of 1 week. During this time, the animals were habituated to head fixation, which was achieved by inserting the head post into a matching head post holder (Luigs and Neumann) located inside a sound-isolated chamber. During habituation sessions (15–30 min), animals learned to lick a reward spout placed in front of their mouth to obtain a liquid reward (10% sucrose) released from an overhead reservoir using a solenoid valve (003-0218-900, Parker Hannifin). Licks were detected using a custom-built infrared emitter and detector on either side of the reward spout. Behavioral events were detected and reward delivery was controlled using a microcontroller (Arduino Uno, Arduino). Following habituation (3–5 days) animals learned to press a lever in order to obtain reward. Reward was delivered if the lever press occurred at least 100 ms following the previous lever release and if the lever was held down for at least 100 ms. These criteria were required to reinforce stereotypical lever pressing behavior and to minimize the reinforcement of accidental lever touches (e.g., during grooming or posture adjustments). The reward amount was gradually decreased over days, to ~3 μL in order to maximize the number of lever presses in each session.

Once animals were lever pressing reliably (>250 lever presses per session, typically 5–7 training days), they underwent a second, brief surgery for making a craniotomy over the auditory cortex. Animals were anesthetized with isoflurane and placed in a stereotaxic frame as described above ("Surgical procedures"). The silicone elastomer was removed from the skull and a small craniotomy was made over the left auditory cortex using a high-precision hand drill (Proxxon, Micromot 50/E) while leaving the dura intact, and sealed again with silicon elastomer. On the following day, animals were head-fixed, the silicone elastomer was removed and a small drop of silicone oil applied to the craniotomy to prevent the brain from drying. A 32-channel (A1X32-Edge-5mm-20-177 or A1X32-Poly2-5mm-50s-177-A32, Neuro-Nexus) or a 64-channel (A1X64-Poly2-6mm-23s-160, NeuroNexus) silicon probe was then inserted into the brain using a micromanipulator (LN-Junior 16, Luigs and Neumann) at a speed of 2 μm/s. Silicon probes were inserted perpendicular to the brain surface (30–35 degrees relative to the horizontal plane) in order to align the electrode sites perpendicular to the layers of the auditory cortex. The silicon probe shank was painted with a lipophilic fluorescent dye (DiI, DiO, or DiD, Life Technologies) to aid in the subsequent identification of the probe location.

After electrodes had been advanced to their final position and following a resting period of 15 min to allow the brain tissue to settle the lever was made available to the animal and neural activity was recorded while the animals pressed the lever as they had been trained to do. Now, however, each lever press triggered the delivery of an auditory stimulus consisting of a white noise burst (100 ms duration, 65 dB SPL) generated by a 24-bit digital-to-analog converter (RZ6, Tucker-Davis Technologies) using RPvdsEx software (Tucker-Davis Technologies) and delivered from a speaker (# R1904/613001, Scan-speak) located 20 cm above and 27 cm to the right of the mouse. Sound intensity for the white noise stimulus was calibrated using a handheld digital sound level meter (CZ18, Colemeter). The same auditory stimulus was also randomly presented from the same speaker throughout the session every 5–10 s, allowing us to measure neural responses to the same physical stimulus when it was either self-generated or randomly generated. The delivery of auditory stimuli was triggered using the same microcontroller that detected behavioral events. Raw neuronal signals were recorded continuously while animals experienced self-generated and random sounds. The signals were filtered between 1 Hz and 7500 Hz, digitized at 30 kHz using a digitizing headstage (RHD2132 Amplifier Board, Intan Technologies) and acquired using an USB interface board (RHD2000, Intan Technologies). Skull screws over the frontal cortex and cerebellum served as ground as reference, respectively. The USB interface board also registered the timestamps of all behavioral events and auditory stimuli from TTL pulses delivered by the microcontroller. In a subset of animals, we examined responses of auditory cortical neurons to randomly presented sounds (white noise burst, 20 ms duration, 65 dB SPL) that were presented in pairs with different inter-pulse intervals (100, 200, 400, 700, and 1000 ms). These recordings were performed immediately after animals had completed the lever-pressing paradigm, while they passively listened to the pairs of sounds with the lever retracted and without moving the recording electrodes. All further analyses of neural and behavioral data were performed offline (see below).

### Recording of neural activity during optogenetic stimulation of M2 axon terminals

Following recovery from surgery for headpost implantation (see "Surgical procedures", above) recordings were performed in the auditory cortex of ChR2-injected animals while optogenetically stimulating axon terminals of M2 neurons. To this end, blue light pulses (473 nm, 100 ms) were delivered to the auditory cortical surface every 2–3 s while recording neuronal activity from the auditory cortex using silicon probes. Light was delivered by a laser (LuxX 473-100, Omicron Laserage) through an optic fiber attached to the silicon probe (diameter: 125 μm; numerical aperture: 0.22; A1X32-10mm-50-177-OA32, Neuronexus) or through a separate optic fiber (diameter: 200 or 400 μm, numerical aperture: 0.37; MFC_200/245-0.37_30mm_SM3_FLT or MFC_400/430-0.37_30mm_SM3_FLT, Doric Lenses) positioned ~1.5 mm above the auditory cortex. In the latter case, recordings were performed using a 64-channel silicon probe (A2X32-5mm-25-200-177, Neuronexus or ASSY-77-H3H3, Cambridge Neurotech). Light intensity was 30 mW when the optic fiber was attached to the silicon probe and 60–65 mW when a separate fiber was used. During these recording sessions, animals sat passively in the head-fixing apparatus but received liquid reward intermittently. Otherwise, recordings were performed as described above (see "Recording of neural activity during self-generation of auditory stimuli in head-fixed mice") except that neural signals were filtered between 500 and 7500 Hz in order to minimize artefacts caused by light stimulation. In a subset of these sessions, we performed separate recordings where we optogenetically stimulated the cell bodies of M2 neurons by placing the optic fiber above M2. To control for the possibility that light might affect neuronal activity independently of ChR2 expression (e.g., through tissue heating), we also examined responses of auditory cortical neurons to blue light pulses delivered to the auditory cortex in mice in which M2 was not transfected with ChR2. In these experiments, light was delivered and neural activity was recorded in the same manner as in mice expressing ChR2 in M2; specifically, light pulses (100 ms, 473 nm, 30 mW) were delivered through an optic fiber (diameter: 125 μm; numerical aperture: 0.22) attached to a silicon probe (A1X32-10mm-50-177-OA32, Neuronexus).

### Histology

Animals were anesthetized with Na-pentobarbital and perfused transcardially with phosphate-buffered saline (pH 7.4) containing 4% paraformaldehyde (PFA) and 15% picric acid. The brain was then removed, stored overnight in 4% PFA and transferred to a 0.01 M PBS solution (10% sucrose, 0.05% NaN₃, pH = 7.4). Brains were then sectioned with a vibratome (VT1000S, Leica) at a thickness of 60 μm. To enhance visualization of eGFP, brain sections were incubated in rabbit anti-GFP

antibody (1:1000; Invitrogen, Catalog #A11122) overnight at room temperature followed by overnight incubation in a secondary anti-rabbit fluorescent antibody (488 nm; 1:750; Invitrogen, Catalog #A11008). Sections were then stained with 4′,6-diaminin-2- pheny-lindol (DAPI) solution (1:5000 in PBS, Molecular Probes, Catalog #D1306) for 5 min, after which they were mounted on microscope slides, dried at room temperature, embedded in a mounting medium (Vectashield, Vector Laboratories) and cover-slipped. Brain sections were subsequently imaged using a confocal laser-scanning microscope (Eclipse90i, Nikon) and acquired using NIS-Elements (Nikon). Each channel of the ND2 file was imported to MATLAB using the Bio-Formats toolbox[73] and saved as a monochrome.tif image for further processing.

## Processing of neuronal data

Raw electrophysiological data were preprocessed by subtracting from each channel the median signal across all functional electrode sites[74]. For recordings performed during self-generation of sounds, each channel was filtered between 300 and 6000 Hz using a 3rd order Butterworth filter. Spike sorting was performed using Klusta (https://github.com/kwikteam/klusta)[75]. Briefly, spikes were detected as local spatiotemporal events using a double-threshold flood fill algorithm (SpikeDetekt) with the strong and weak thresholds set to 4.5–5 and 2–2.5 times the standard deviation of each channel, respectively. The first three principal components of the waveforms of detected spikes on each channel were then used to cluster the spikes using an auto-mated masked expectation-maximization algorithm (Masked Klus-taKwik). This was followed by manual refinement of the clusters based on visual inspection of their spike waveforms as well as their auto- and cross-correlograms using phy Kwik GUI. Following spike sorting, we computed for each neuron its average spike waveform (495 μs before to 1287 μs after waveform trough) from the channel on which the waveform was largest. The spike half-width (trough width at half minimum) and trough-to-peak separation of each neurons' waveform were then used to separate neurons into putative pyramidal neurons and interneurons. To this end, the distributions of these two waveform features were fit using a 2-dimensional Gaussian mixture model[76]. Neurons with low classification confidence ($p < 0.95$ of belonging to the assigned class) were excluded from analyses comparing putative pyramidal neurons and interneurons. The depth of each recorded neuron was estimated from the position of the electrode site on which its waveform was largest and the depth of the silicon probe tip below the brain surface, estimated from the micromanipulator travel distance.

## Analysis of evoked responses

In order to examine sound-evoked responses, we computed peri-stimulus time histograms (PSTHs) aligned around the onset of random and self-generated sounds. PSTHs were calculated from a subset of these sounds in order to minimize the influence of two confounding variables that can influence sensory responsiveness of auditory cor-tical neurons. First, in order to minimize the influence of sensory adaptation[40], we included only random and self-generated sounds that occurred at least 1 s following the previous sound, but similar results were obtained using other time cutoffs (Supplementary Fig. 2A). Sec-ond, in order to minimize differences in overall behavioral state during the occurrence of random and self-generated sounds, we excluded random sounds if the sound preceding it was also randomly generated. The reason for this is that the animals would sometimes generate sounds (lever-press) in bouts and then stop for several minutes before resuming lever pressing again. In the time periods between such lever pressing bouts the animals might be in a more general state of beha-vioral quiescence (e.g., drowsiness) which on its own can have a large effect on auditory responsiveness when compared to states of beha-vioral activity[9,37,77] as for example during lever-pressing bouts. For this

reason, we excluded random sounds from analysis that occurred outside lever-pressing bouts. For detecting these random sounds, we reasoned that the sound preceding them should be a random sound, not a self-generated one, which was therefore used as the criterion for their exclusion. This selection procedure is the same as what we used previously when studying responses to self-generated sounds in wild-type mice (for an illustration of this selection procedure, see Fig. 1B in ref. 9). However, using alternative methods for selecting randomly-generated sounds for analysis did not alter our main results (Supple-mentary Fig. 2B, C).

We then quantified the response amplitude as the average firing rate between 10 and 50 ms following stimulus onset minus the baseline firing rate (0–200 ms before stimulus onset). The 10–50 ms response window was chosen based on the short latency and brief duration of the responses of most neurons (see Fig. 1c, d). In order to classify neurons as auditory responsive, we divided the response amplitude by the standard deviation of firing rates measured in 5 ms bins during the 200 ms period preceding stimulus onset. Neurons were considered auditory responsive and included for further analysis if their response amplitude following either random or self-generated sounds was at least two times greater than the standard deviation of the baseline period. In order to minimize the effects of sensory adaptation, sounds were excluded from analysis that occurred less than 1 s after the pre-vious sound. Only random sounds occurring during periods of lever pressing were included in analyses; specifically, if they were preceded by a self-generated sound. This was done to minimize differences in behavioral state between the two types of sounds. To quantify the differences in responses to self-generated and random sounds for each neuron, we computed a modulation index (MI) by subtracting its mean response to random sounds from its mean response to self-generated sounds and dividing this difference by the sum of the two responses:

$$\mathrm{MI} = [\mathrm{Response}_{\mathrm{Self-generated}} - \mathrm{Response}_{\mathrm{Random}}]/[\mathrm{Response}_{\mathrm{Self-generated}} + \mathrm{Response}_{\mathrm{Random}}]$$

We also examined whether each neuron responded significantly differently to self-generated and random sounds by calculating its responses to each sound presentation and comparing responses to random and self-generated sounds using the rank-sum test. In order to compare responses during early and late blocks of self-generated sounds, we calculated the MI separately for the first 300 self-generated sounds (early block) and the subsequent 300 self-generated sounds (late block; or the remaining sounds in sessions with less than 600). In each block, random sounds occurring within the block were used to calculate the MI. Sessions with less than 400 self-generated sounds in total, or less than 20 self-generated or random sounds meeting criteria for inclusion (see above) in either block, were excluded from analysis. This led to the exclusion of two animals from each genotype. In order to minimize the number of excluded sessions, we excluded only sounds from analysis that occurred less than 0.5 s following the pre-vious sound, but similar results (with fewer animals) were obtained using a 1 s threshold as in our other analyses. For examining motor preparatory activity in auditory cortical neurons (Fig. 3), we analyzed neural activity preceding lever presses that occurred at least 2 s fol-lowing the previous sound (self-generated or random), in order to eliminate the influence of previous sounds and lever pressing behavior on neuronal activity. In order to examine responses to pairs of random sounds (Supplementary Fig. 3) we calculated a paired-pulse ratio, defined as the response to the second sound in a pair, divided by the response to the first sound (average response 10–50 ms after stimulus onset). Inter-pulse intervals of 100 ms were excluded from analysis since responses to the first sound sometimes extended beyond 100 ms.

To examine responses of neurons to optogenetic stimulation of M2 axon terminals, we first calculated for each cell its average PSTH

from 100 ms before to 100 ms after light onset in 5 ms bins. We then quantified for each bin following light onset the difference in its firing rate ($\Delta$FR) relative to baseline (0–100 ms before light onset). In order to assess the significance of $\Delta$FR values we created surrogate PSTHs with the same duration and bin size by selecting 200 ms segments randomly from the baseline periods preceding light onset. The number of selected segments corresponded to the number of light presentations used to generate the actual PSTH. The $\Delta$FR was then calculated for each 5 ms bin in the second 100 ms of the surrogate PSTH relative to the average rate in the first 100 ms. This was repeated 1000 times, yielding a distribution of $\Delta$FR values expected by chance. The distribution was then used to calculate the $p$ value of the $\Delta$FR values in each bin of the actual PSTH, defined as the fraction of bins of the surrogate PSTHs with higher or lower $\Delta$FR values, depending on whether the $\Delta$FR in the actual PSTH was positive or negative, respectively. The latency of excitatory responses was defined as the first 5 ms bin whose $\Delta$FR had a $p$ value of <0.001. Cells were classified as having short-latency excitatory responses if their response latency was less than or equal to 10 ms (or 15 ms in the subset of recordings where the cell bodies of M2 neurons were optogenetically stimulated) and if they emitted at least ten spikes in that bin across light presentations. Cells were classified as having inhibitory responses if they had at least three consecutive bins with negative $\Delta$FR values passing a significance threshold of $p < 0.05$ during the first 50 ms following light onset and if they emitted at least 20 spikes in the baseline period across trials. The spike number thresholds were used to avoid false positives due to low spike counts.

### Analysis of retrogradely labeled neurons in frontal cortex

In order to visualize and quantify eGFP-expressing frontal cortical neurons retrogradely labeled from the auditory cortex in $Df(16)A^{+/-}$ mice and their wild-type littermates (see "Surgical procedures", above), 5 coronal brain sections were imaged in each animal at anteroposterior positions corresponding approximately to 2.46, mm 1.94 mm, 1.42 mm, 0.89 mm and 0.39 mm anterior to bregma, according to ref. 78. Each section was imaged in its entirety using a confocal microscope at 10× magnification and at 12 different focal planes (separated by 7.025 μm). The maximum intensity projection (MIP) of the resulting z-stack was calculated and used for subsequent analysis. To examine retrogradely labeled neurons in the auditory cortex contralateral to the injection site, we imaged the brain section corresponding to the injection site in the same way (at nine focal planes with 8.57 μm separation) and used the MIP for analysis. To quantify expression in the auditory thalamus, the section with the highest thalamic expression was imaged at 20× at six focal planes (separated by 10 μm) and the MIP was used for subsequent analysis.

Retrogradely labeled (eGFP-expressing) cells were detected using a semi-automatic analysis pipeline within ImageJ (National Institute of Health). First, each image used for analysis was converted to 8-bit monochrome and pre-processed to reduce background ('Subtract Background' in ImageJ; rolling ball radius = 30 pixels) and enhance contrast ("Enhance Contrast" in ImageJ; % saturated pixels = 0.5) followed by median filtering (1 pixel radius). The Trainable Weka Segmentation plugin[79] was then used to train a classifier that assigned each pixel in the image as belonging to either a cell or background. Separate classifiers were trained for sections including frontal cortical areas, the auditory cortex and the auditory thalamus. The classified binary image was then further processed in ImageJ using a morphological open operator and the watershed algorithm. An output file was then created specifying the X and Y position of each detected cell within the image ("Analyze particles" in ImageJ; minimum size: 25 μm). The number of cells detected using these steps was similar to numbers obtained by manual cell counting, as estimated in a subset of brain sections (Supplementary Fig. 5D).

In order to count the number of cells in each brain region, we aligned reference atlas templates of the the Allen Mouse Brain Common Coordinate Framework (CCFv3[80]) to each of the analyzed brain sections semi-automatically using the WholeBrain software (http://www.wholebrainsoftware.org/)[81] implemented within the SMART software package[82]. Briefly, for each brain section, the appropriate coronal template from the reference atlas was chosen. This template was initially fit automatically to the brain section using 32 points arranged along the surface of the brain. The fit was subsequently improved in an iterative fashion by manually adding additional points at corresponding locations in the template and the image and repeating the fitting procedure. In total 60–100 points were used to create the final fit. The resulting information about the boundaries of each brain region was then used to assign each cell to a specific brain region, hemisphere and cortical layer based on its X and Y position within the image. For each animal, the number of retrogradely labeled neurons in each region was combined across all imaged sections.

### Statistics

Statistical differences between means were determined using the Wilcoxon signed rank test, Wilcoxon ranked-sum test, one-way and two-way ANOVA, as described in the text. Statistical differences between ratios were determined using Fisher's exact test. A $p$ value of <0.05 was used as the criterion for statistical significance, unless otherwise noted. All tests were two-sided and corrections were not performed for multiple comparisons Summary statistics are reported in the text as the mean ± s.e.m. In the figures, data is summarized as the mean (where individual data points are shown), the mean ± s.e.m or with box plots representing the median (line), 25th and 75th percentiles (box), and 5th and 95th percentiles (whiskers). For analyses of electrophysiological data, statistical differences were calculated using all neurons combined across animals within each genotype. However, since this might violate the assumption of independence of samples and cause biased estimates of statistical significance, we repeated our key analyses using hierarchical bootstrapping[83,84] which avoids this bias. Briefly, bootstrapping samples were generated by resampling with replacement, first from animals within a genotype and then within neurons from the resampled animals. This was repeated 1000 times for each genotype, each time calculating a bootstrap mean of a variable of interest (e.g., the modulation index). Significance was then defined as the proportion of bootstrap samples where the bootstrap means of the two genotypes were either the same or differed in the opposite direction to the reported mean genotype difference.

### Reporting summary

Further information on research design is available in the Nature Portfolio Reporting Summary linked to this article.

## Data availability

Source data are provided in this paper. Raw data are available from the corresponding author upon request. Source data are provided in this paper.

## Code availability

Code for generating the main figures from the source data is available on https://github.com/torfisigurdsson/22q11CDPaper (https://doi.org/10.5281/zenodo.8414756).

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

## Acknowledgements

This work was supported by the German Research Foundation (DFG) grant SI 1942/2-2 to T.S. We would like to thank Sebastian Betz, Beatrice Fischer, Jasmine Sonntag and Thomas Wulf for technical assistance and Sevil Duvarci for help with illustrations and for comments on the manuscript.

## Author contributions

T.S., B.P.R., S.B., and S.S.B. designed the experiments. B.P.R., S.B., and S.S.B. performed the experiments. B.P.R., S.B., T.S., and S.S.B. analyzed the data. J.A.G. provided resources and conceptual input. T.S. wrote the paper with input from all authors. T.S. supervised the study and obtained funding.

## Funding

## Competing interests

The authors declare no competing interests.
