## [Peer Review File · Nature Communications]

Altered corollary discharge signaling in the auditory cortex of a mouse model of schizophrenia predispositionREVIEWER COMMENTS

Reviewer #1 (Remarks to the Author):

The Df(16)A/+ mouse line is an animal model for one of the largest genetic risk factors for human schizophrenia, the 22q11.2 microdeletion. This manuscript shows that these mice also model a neurophysiological endophenotype in schizophrenia: reduced attenuation of brain responses to self-generated sounds. The authors also provide compelling evidence from neurophysiological, anatomical tracing, and functional connectivity experiments suggesting that the reduced attenuation of auditory cortical responses arises from a reduction in the strength of motor cortex input to the auditory cortex.

This manuscript includes several important and novel findings:

(1) There is less suppression of auditory cortical responses to self-generated sounds (relative to random sounds) in Df(16)A/+ mice than in their WT littermates. This result (Figure 1) is very clean and compelling, because auditory cortical responses to random sounds are statistically indistinguishable between the animal groups. Also, the authors show nicely that behavioural performance of the lever pressing task is no different in Df(16)A/+ and WT mice.

(2) Reduction in suppression of cortical responses to self-generated sounds in Df(16)A/+ mice is observed in both pyramidal cells and putative inhibitory interneurons, and in all layers of the cortex, although the abnormality appears most pronounced in deep cortical layers.

(3) Motor preparatory activity in the auditory cortex is reduced in Df(16)A/+ mice relative to their WT littermates, suggesting an abnormality in motor inputs to the auditory cortex.

(4) Motor cortex inputs to the auditory cortex are weaker in Df(16)A/+ mice than in their WT littermates.

The manuscript uses both retrograde tracing and optogenetic techniques to

demonstrate that motor areas M1 and M2 (and orbitofrontal cortex) project anatomically and functionally to the auditory cortex, eliciting likely monosynaptic responses in deep cortical layers and polysynaptic responses in all cortical layers. The authors also show this connectivity influences auditory cortical activity. These functional connectivity findings are not novel and have been shown previously before in WT mice, but provide useful context for interpretation of results in Df(16)A/+ mice, since the Df(16)A/+ abnormality in attenuation of evoked responses to self-generated sounds is most pronounced in the deep layers of the auditory cortex.

The conclusion of this study is that functional projections from the motor cortex to the auditory cortex are weaker in Df(16)A/+ mice than in WT animals, and this weakening of motor-cortex-to-auditory-cortex projections leads to reduced suppression of auditory cortical responses to self-generated sounds. More speculatively, the findings lend support to the theory that reduced corollary discharge might contribute to hallucinations and delusions of control in schizophrenia, e.g. through misattribution of self-generated sounds and movements to external sources.

The data in this paper is very compelling and the authors carefully address many possible confounds, or are able to rule them out because of statistical similarities in responses to random sounds in the Df(16)A/+ and WT groups. The manuscript is also very well-written, with well-chosen and well-presented figures. I have only two suggestions, or points for possible additional discussion.

(1) The manuscript explains that only responses to random sounds that occurred during a lever press were compared to responses to self-generated sounds. Additionally, to be included in the analysis, the random sounds were required to have occurred at least 0.5 sec after a self-generated sound. The authors also mention that they re-did the analysis using only sounds occurring at least 1 sec after a self-generated sound and obtained similar results.

It would be helpful to discuss at greater length in the manuscript why these particular "random" sounds were chosen for comparison with self-generated sounds. Presumably the

reasoning here is that somatosensory input will be the same for self-generated and random sounds if both have occurred during a lever press, and the random sound must therefore come after the self-generated one because (of course) the self-generated sound would occur immediately after lever press initiation.

However I wonder about the fact that the random sound is always happening *after* the self-generated one. Is it possible that there could be differences between Df(16)A/+ and WT mice in recovery of auditory cortical activity following sound input, whether self-generated or not, which contribute to the observed abnormality? The figures show only measures of firing rate to the sounds, not the duration of the responses. Were there differences between Df(16)A/+ and WT mice in the temporal characteristics of auditory cortical responses to self-generated sounds or random sounds? If so, could there be differences in interaction between self-generated and random sound responses over a 0.5-1 sec timescale?

In the methods text there is a mention of comparisons to "random" sounds occurring at least 2 sec before lever press, but I didn't see where these data were shown or discussed in the main text. It might be useful to have an additional supplementary figure demonstrating that the key difference in response attenuation between genotypes is evident both for random sounds occurring >1 sec after the self-generated sound --- and, if possible, for random sounds occurring well before the lever press. (This may not be possible to show, I understand, because random-sound responses might be very noisy in the latter case when the animal's behaviour is less well controlled than during a lever press.)

(2) The issue of hearing impairment in 22q11.2DS patients and at least some mouse models of 22q11.2DS is acknowledged in the Discussion, but discounted mostly on the grounds that it has not been demonstrated in all mouse models of 22q11.2DS and may be due to differences in laboratory conditions. True, but in fact the data in the manuscript provide a much stronger argument that hearing impairment would be an unlikely explanation for the effects observed here.

Hearing thresholds are essentially bimodal in both 22q11.2DS patients and the one mouse

model of 22q11.2DS where they have been intensively investigated (i.e., some individuals have normal hearing and others have mild to moderate impairment). There is no evidence for bimodality in the Df(16)A/+ data here. Moreover, the magnitude of evoked responses to random sounds appear to be statistically indistinguishable between the Df(16)A/+ and WT mice, and there doesn't seem to be any obvious difference in the degree of self-generated sound attenuation depending on the magnitude of the response evoked by random sounds (i.e., there's no apparent nonlinearity in the Figure 1 E scatterplots), as might be expected if hearing impairment were a confound. Were spontaneous firing rates statistically indistinguishable in Df(16)A/+ and WT mice (as suggested already by Figure 3 B)? If so then there's no evidence for a difference in baseline excitability of the cells either. With this degree of similarity between Df(16)A/+ and WT mice in evoked responses to random sounds (and spontaneous rates?), it seems likely that the key conclusions regarding attenuation of self-generated sounds would be robust to any variability in hearing thresholds among Df(16)A/+ mice even if it were present.

To be clear, I am not suggesting that the authors need to do additional analyses to address this point, only that they could make a stronger case in the Discussion that their conclusions are highly unlikely to have been confounded by hearing impairment, even if it does occur in a subset of the Df(16)1/+ animals.

Reviewer #2 (Remarks to the Author):

I enjoyed reading "Impaired corollary discharge signaling in the auditory cortex of a mouse model of schizophrenia predisposition" from the Sigurdsson lab. Even ancient philosophers knew that there had to be a mechanism such as the corollary discharge that enables animals to know whether sensations they are experiencing result from their own actions or from those of an approaching predator. Recently, clever investigators like these have begun to help the rest of us understand the neurobiology of this mechanism. This paper represents a step forward in the basic science of the corollary discharge, while also linking the dysfunction of this mechanism to a mouse model of schizophrenia related to the 22q.11 deletion.

I have very few suggestions to improve this paper, but here are a couple.

They say:

“Disruptions in the filtering of self-generated stimuli may also contribute to the symptoms of psychiatric disease.” They should temper this by saying instead: “Disruptions in the filtering of self-generated stimuli have been reported across the psychosis spectrum.”

They say:

“Such a failure to predict the sensory consequences of action could cause self-generated stimuli to be misattributed to an external source, leading to the hallucinations and delusions that are characteristic of schizophrenia (Feinberg and Guazzelli, 1999; Allen et al., 2007; Jones and Fernyhough, 2007; Fletcher and Frith, 2009; Thakkar et al., 2021).” Most, if not all, of these papers are reviews. They might consider adding another list of empirical papers that report a relationship between hallucinations/delusions and deficits in this system. They should search for Bonsal, Thakkar, Shergill, and Ford lab papers, among others.

I am not aware of these type studies being done in humans with the 22q11.2 microdeletion. If there are some, the authors need to mention them. If there are not, they should mention that too.

They cite the Singla et al Nature Neuroscience paper on a cerebellum-like circuit in the auditory system cancels responses to self-generated sounds. Can the authors say anything about the role of the cerebellum in this study? Did they not look in the cerebellum, or did they not see a signal in the cerebellum?

Thank you for your excellent work,

Judith M. Ford

Reviewer #3 (Remarks to the Author):

This study investigates the distinction between self- and externally-generated stimuli in the Df16 mouse model of schizophrenia. The main finding is that Df16 vs WT mice show

decreased attenuation of single neuron responses to self-generated sounds in auditory cortex. The researchers further show decreased preparatory motor activity in motor cortex and decreased numbers of motor-auditory projections in Df16 vs WT mice. They also establish that motor-auditory projections affect single neuron activity in auditory cortex in WT mice. The researchers conclude that decreased top-down signalling from motor cortex leads to a blurred distinction between self- and externally-generated stimuli in auditory cortex.

This is an elegant and original study that addresses the relevant question of how neural circuit dysfunction gives rise to sensory processing deficits relevant to schizophrenia. Reduced distinction between self and non-self has long been related to hallucinations and delusions in schizophrenia. This study now suggests a potential neural circuit mechanism for the well-established macro-level alterations in schizophrenia patients. The manuscript is well-written and the methodology is sound and described in sufficient detail to allow other researchers to reproduce the work. I have some concerns regarding the statistics and the interpretation of the results that I outline below. If adequately addressed, this work would make an extremely valuable contribution to both psychiatric and systems neuroscience.

Comments:

1. In their statistical analyses, the researchers pool neurons across subjects. Although this practice has long been standard practice in the field, it is increasingly recognised that pooling dependent and independent observations violates the assumption of most statistical tests and can lead to biased statistical inferences (e.g., see here for a recent discussion <https://www.ncbi.nlm.nih.gov/pmc/articles/PMC7906290/>). Therefore, the authors might want to consider validating their key results with statistical tests that are designed to accurately deal with hierarchical data sets (e.g., linear mixed models, hierarchical bootstrap). At the very least, the researchers could show how consistent the results are across subjects (e.g., by presenting data grouped by subject).

2. It remains open whether the findings of altered neural responses and anatomy have any

functional relevance. The researchers did not assess whether the increased neural responses to self-generated stimuli were paralleled by changed behavioural responses. While this does not necessarily detract from the value of these findings, it seems a stretch to claim that the study establishes any “impaired” signalling, as this implies an impact on function. The authors could consider providing some additional results on the behavioral consequences of the altered neural responses. Alternatively, the authors could use more descriptive terms when talking about their results and discuss how the functional aspect could be considered in the future.

3. Relatedly, the optogenetics experiments show that activity in motor cortex-auditory cortex projections has an impact on activity in auditory cortex. However, because these experiments are confined to wild-type animals, it remains open whether the motor-auditory projections function in the same way in Df16 and in WT mice, and whether activation of these projections during behaviour could even rescue the attenuation of responses to self-generated stimuli in auditory cortex. Again, this does not necessarily detract from the value of the study, as the authors already provide two neural circuit explanations for the observed auditory cortex responses (decreased preparatory activity in motor cortex AND decreased number of projections). However, it seems misleading to claim “weakened” projections when the functional strength of these projections has not been assessed in Df16 mice, and the authors could consider using more descriptive words when presenting and discussing their results.

4. The researchers show that light delivery to AudCx with ChR2-expressing M2 terminals leads to responses in some neurons. Can the researchers be sure that these responses are indeed mediated by the activation of M2 terminals, or might the observed responses be a direct effect of the light on the neurons? It seems that the experiments using light delivery to M2 (Fig. S2) provide some support for the first interpretation. The authors might want to consider being more explicit about this and/or consider additional arguments for their interpretation of the optogenetics results.

Minor suggestions:

Page 5, line 27 “and suggests that this learning is impaired in Df(16)A+/- mice” -> This conclusion does not seem justified as attenuation is decreased both at early and late timepoints and there is no interaction between genotype and timepoint which would suggest a difference in learning.

Page 5, line 30 “reveal a cellular basis for similar deficits seen in schizophrenia patients” -> This conclusion does not seem justified by the data, as this is a study in a mouse model of a genetic syndrome associated with a high risk for schizophrenia without any patients tested. The authors might want to consider using some modifiers (“could”, “potential”, etc.).

Page 7, line 1: What test does the p-value refer to?

Figures captions: How many mice were used in each plot?

Figure 1/3/6: “Impaired” and “weaker” imply an impact to function, but the figures depict decreases in neuron activity/numbers which might or might not have an impact on function. The researcher could consider using more descriptive terms. This might seem like semantics but words can matter when presenting results related to a condition that is affecting people who might read or hear about this work.

Figure 5, caption: “Percentages are relative to the total number of neurons in each brain region” -> “Percentages are relative to the total number of labelled neurons in each brain region” (I assume as the numbers seem to add up to 100).

We would like to thank the reviewers for their overall positive assessment of the manuscript and the many constructive comments which have helped us improve it. Below we provide a point-by-point response to each comment. We would also like to point out a few additional minor changes and corrections we have made to the manuscript:

1. In Figure 6B, we realized that the cells labeled in the contralateral auditory cortex were not easily visible. Furthermore, a delineation of the different auditory cortex subregions was missing. We have therefore cropped this figure around the auditory cortex and enlarged it and also added the boundaries for A1, DA and VA.
2. We realized that an error in our code had caused some of the neurons recorded from ChR2-expressing mice to be included twice in the dataset, thus slightly inflating the total number of neurons reported in Figure 4 (to 913 instead of 886). This has now been corrected in the revised manuscript but has not altered the main results (i.e. the ratios of excited and inhibited neurons).
3. In Figures 1E-F and 3C in the original manuscript, the scaling of the axes, which was chosen to best visualize the spread of the data, led to the exclusion of several data points that were outside the axis boundaries. This has now been indicated in the corresponding figure legends. We also have added descriptions of display elements (box plots, lines indicating means) that had not been defined in the original manuscript.
4. In Figure 3C, we wrote that the analysis window was 0-100 ms before the lever press. However, we realized that the figure as well as the statistics reported in the text were based on an analysis window of 0-200 ms in the text. We have left this analysis window as is but corrected its description in the text.

Reviewer #1 (Remarks to the Author):

The Df(16)A/+ mouse line is an animal model for one of the largest genetic risk factors for human schizophrenia, the 22q11.2 microdeletion. This manuscript shows that these mice also model a neurophysiological endophenotype in schizophrenia: reduced attenuation of brain responses to self-generated sounds. The authors also provide compelling evidence from neurophysiological, anatomical tracing, and functional connectivity experiments suggesting that the reduced attenuation of auditory cortical responses arises from a reduction in the strength of motor cortex input to the auditory cortex.

This manuscript includes several important and novel findings:

(1) There is less suppression of auditory cortical responses to self-generated sounds (relative to random sounds) in Df(16)A/+ mice than in their WT littermates. This result (Figure 1) is very clean and compelling, because auditory cortical responses to random sounds are statistically indistinguishable between the animal groups. Also, the authors show nicely that behavioural performance of the lever pressing task is no different in Df(16)A/+ and WT mice.

(2) Reduction in suppression of cortical responses to self-generated sounds in Df(16)A/+ mice is observed in both pyramidal cells and putative inhibitory interneurons, and in all layers of the cortex, although the abnormality appears most pronounced in deep cortical layers.

(3) Motor preparatory activity in the auditory cortex is reduced in Df(16)A/+ mice relative to their WT littermates, suggesting an abnormality in motor inputs to the auditory cortex.

(4) Motor cortex inputs to the auditory cortex are weaker in Df(16)A/+ mice than in their WT littermates.

The manuscript uses both retrograde tracing and optogenetic techniques to demonstrate that motor areas M1 and M2 (and orbitofrontal cortex) project anatomically and functionally to the auditory cortex, eliciting likely monosynaptic responses in deep cortical layers and polysynaptic responses in all cortical layers. The authors also show this connectivity influences auditory cortical activity. These functional connectivity findings are not novel and have been shown previously before in WT mice, but provide useful context for interpretation of results in Df(16)A/+ mice, since the Df(16)A/+ abnormality in attenuation of evoked responses to self-generated sounds is most pronounced in the deep layers of the auditory cortex.

The conclusion of this study is that functional projections from the motor cortex to the auditory cortex are weaker in Df(16)A/+ mice than in WT animals, and this weakening of motor-cortex-to-auditory-cortex projections leads to reduced suppression of auditory cortical responses to self-generated sounds. More speculatively, the findings lend support to the theory that reduced corollary discharge might contribute to hallucinations and delusions of control in schizophrenia, e.g. through misattribution of self-generated sounds and movements to external sources.

The data in this paper is very compelling and the authors carefully address many possible confounds, or are able to rule them out because of statistical similarities in responses to random sounds in the Df(16)A/+ and WT groups. The manuscript is also very well-written, with well-chosen and well-presented figures. I have only two suggestions, or points for possible additional discussion.

We thank the reviewer for the careful reading of our manuscript and for the many thoughtful and constructive comments.

(1) The manuscript explains that only responses to random sounds that occurred during a lever press were compared to responses to self-generated sounds. Additionally, to be included in the analysis, the random sounds were required to have occurred at least 0.5 sec after a self-generated sound. The authors also mention that they re-did the analysis

using only sounds occurring at least 1 sec after a self-generated sound and obtained similar results.

It would be helpful to discuss at greater length in the manuscript why these particular "random" sounds were chosen for comparison with self-generated sounds. Presumably the reasoning here is that somatosensory input will be the same for self-generated and random sounds if both have occurred during a lever press, and the random sound must therefore come after the self-generated one because (of course) the self-generated sound would occur immediately after lever press initiation.

We thank the reviewer for drawing our attention to this important aspect of our methods, which we now realize could have been explained more clearly and in more detail in order to avoid misunderstanding. Specifically, we now see that our statement "Only random sounds occurring during periods of lever pressing were included in analyses" was unfortunate since it can be taken to mean that random sounds were selected that occurred while the lever was being pressed (held down), which was in fact not how they were selected. Below, we describe in detail our selection procedure and the reasoning behind it, as well as how we have improved the manuscript in order to make this aspect of our methods more clear to the reader.

As the reviewer correctly points out, responses to only a subset of random sounds were included for analysis; it is also important to note that a similar selection procedure was applied to the self-generated sounds. This selection procedure was intended to minimize the influence of two confounding variables that could have influenced neuronal responses to the sounds, independently of whether they were randomly generated or self-generated. The first confound is sensory adaptation: in the auditory system, sounds will typically elicit smaller responses if they are shortly preceded by another sound (e.g. Wehr and Zador, 2005). This could happen in our paradigm by chance (e.g. a random sound occurring right after a self-generated sound) or due to the animal's behavior (e.g. when several sounds are generated in rapid succession by lever pressing). In the methods we wrote "In order to minimize the effects of sensory adaptation, sounds were excluded from analysis that occurred less than 1 seconds after the previous sound." Importantly, this selection criterion was applied to both random and self-generated sound, which we have now made explicit in the methods (p. 25). This 1 second threshold was used for all analyses except those in Figure 1I and 1J, where we used a 0.5 sec threshold in order to be able to include more animals in the analysis; this we explained in the methods, and also noted that similar results were obtained using a 1 second threshold (p. 25, line 31 in revised manuscript).

The second confounding variable whose influence on auditory responses we wanted to minimize was possible differences in the overall behavioral state of the animals during the delivery of random and self-generated sounds. Although random sounds were presented every 5-10 seconds throughout the recording session, self-generated sounds were produced somewhat more irregularly since they were dependent on the animal's behavior. Specifically, mice would often lever-press in 'bouts' and then stop for several minutes before resuming lever pressing again. In the time periods between such lever pressing bouts the animals might be in a more general state of behavioral quiescence (e.g. drowsiness) which on its own can have a large effect on auditory responsiveness when compared to states of behavioral activity (Schneider et al.,

2014; Zhou et al., 2014; Rummell et al., 2016) as for example during lever-pressing bouts. For this reason, we felt it was important to exclude from analysis random sounds occurring outside lever-pressing bouts. We reasoned that if a random sound occurred outside lever-pressing bouts, the sound preceding it should be a random sound, not a self-generated one. Therefore, a random sound was selected for analysis if the preceding sound was a self-generated one, which indicated that the random sound had occurred during or close in time to a lever-pressing bout, meaning that it occurred during a similar behavioral state as the self-generated sounds. (Note that, as discussed below, in response to another comment from the reviewer we have examined how our results are affected by selecting random sounds occurring shortly before a lever press, when the behavioral state is likely to be even more similar between the two sound types.)

Unfortunately, we did not use the term 'lever-pressing bout' when describing this selection procedure in the methods but rather wrote "Only random sounds occurring during periods of lever pressing were included in analyses". We now realize that 'periods of lever pressing' is a misleading phrase since it can be understood to mean (as the reviewer has, understandably) that only random sounds occurring while the lever was being pressed (i.e. held down) were included in the analyses. However, this was not a requirement for their inclusion, only that they were preceded in time by a self-generated sound and not a randomly-generated sound. That being said, the reviewer is correct that random sounds occurring while the lever is being held down will be associated with similar somatosensory input as self-generated sounds; thus it might be more appropriate to select these random sounds for analysis. We have therefore repeated the analyses shown in Figure 1G after applying this additional selection criterion to the random sounds and found that this did not change the results appreciably (Supplementary Figure 2C in the revised manuscript).

In the revised version of the manuscript we have now described the sound selection procedure and its rationale in more detail in both the Methods (p. 25) and the results (p. 5), incorporating the points mentioned above. We also wish to point out that the same selection procedure was used in our previous study examining the processing of self-generated sounds in wild-type mice, which we illustrated visually with examples of behavioral data (see Figure 1B in Rummell et al., 2016). This figure has now been correspondingly cited in the methods in order to further aid the understanding of our selection procedure. We therefore hope this important aspect of our methods will now be more clear to the reader and thank the reviewer once more for drawing our attention to this important issue.

However I wonder about the fact that the random sound is always happening *after* the self-generated one. Is it possible that there could be differences between Df(16)A/+ and WT mice in recovery of auditory cortical activity following sound input, whether self-generated or not, which contribute to the observed abnormality?

The reviewer raises a valid concern. As discussed above, since random and self-generated sounds can occur close to each other in time, some adaptation of auditory responses to these sounds is expected to occur in our paradigm. Indeed, this is the reason we excluded sounds from analysis (random or self-generated) that occurred less than 1s following a previous sound (or 0.5s in one analysis), as described above. However, although this selection procedure likely minimized

the effects of sensory adaptation, it may not have eliminated it. Furthermore, the strength of adaptation might differ between the two genotypes in terms of its magnitude or time course (we assume this is what the reviewer means by “differences between *Df(16)A^{+/-}* and WT mice in recovery of auditory cortical activity following sound input”). Although it is not obvious how such a difference would have affected our results (since presumably responses to both self-generated and random sounds are influenced by sensory adaptation), we acknowledge that this is a possibility which must be considered.

A first step towards addressing this would be to examine whether any differences in sensory adaptation do in fact exist between the two genotypes. This is difficult to examine with the data collected during our behavioral paradigm since the timing between the sounds was not systematically controlled. However, in a subset of the animals used in the experiments, we did in fact examine responses to pairs of randomly presented sounds separated by different inter-stimulus intervals (ISIs), similar to previous studies (Wehr and Zador, 2005). These data were recorded separately (with the silicon probe left in the same position) after mice had completed the behavioral paradigm where they experienced self-generated and random sounds. We originally planned to include these data in a separate study, but since they are highly relevant to the issue raised by the reviewer, we have now analyzed them and present the results in Supplementary Figure 3 of the revised manuscript. As expected from previous findings (Wehr and Zador, 2005), auditory cortical neurons showed weaker responses to the second sound in a pair; this adaptation was strongest at the shortest ISI examined and decreased gradually with increasing ISIs, demonstrating “recovery” from previous sound stimulation. However, the magnitude and time course of this adaptation was virtually identical in the two genotypes. These results therefore argue against the possibility that differences in the recovery of activity following previous sound input are responsible for the altered processing of self-generated sounds in the *Df(16)A^{+/-}* mice. Also relevant to this issue are the results of analyses performed in response to another suggestion from the reviewer, which show that including random and self-generated sounds for analysis that occur at least 2s after the previous sound, which would further reduce the influence of sensory adaptation, does not alter our main findings (see below). We thank the reviewer for raising this issue and hope that our new analyses have adequately addressed it.

The figures show only measures of firing rate to the sounds, not the duration of the responses. Were there differences between *Df(16)A/+* and WT mice in the temporal characteristics of auditory cortical responses to self-generated sounds or random sounds? If so, could there be differences in interaction between self-generated and random sound responses over a 0.5-1 sec timescale?

This is a good question. In theory, it is possible that ongoing responses to one sound could affect the responses to a subsequent sound (we assume this is what the reviewer means by ‘interaction’). This could for example happen if the first sound causes long-lasting inhibition. However, we think this is unlikely to have influenced our results because, as discussed above, we excluded sounds occurring less than 1s (or in one analysis less than 0.5 s) after the previous sound and also because the auditory responses we observed were relatively transient in nature.

To illustrate this, we have plotted for the reviewer the average peri-stimulus time histograms (PSTHs) to the random (left) and self-generated sounds (right) in the two genotypes, shown below. As can be seen, the primary sound-evoked responses occurred 10-50 ms following stimulus onset, which is the time window we used for quantifying the evoked firing rates (shaded area). A smaller second peak is also apparent in the PSTHs that begins ~110 ms after stimulus onset, which we believe is likely a response to the sound offset (note that the sound duration was 100 ms). However, because this offset response was variable and not straightforward to quantify (since its amplitude partly depends on the response at sound onset), we did not analyze it further. At any rate, after this second peak the PSTHs rapidly decrease back to baseline and 1 second following stimulus onset, which is the earliest time the next sound can occur for it to be included in analysis, there is hardly any response to speak of. The temporal profiles of the PSTHs in the two genotypes also appear qualitatively similar. Thus, we think it unlikely that there is an 'interaction' in this manner between self-generated and random sounds, or between self-generated sounds themselves, that could have influenced our results. Of course, sounds can influence each other via synaptic depression, but this form of interaction does not appear to differ between the genotypes, as we have shown in the revised manuscript (see above and Supplementary Figure 3).

In the methods text there is a mention of comparisons to "random" sounds occurring at least 2 sec before lever press, but I didn't see where these data were shown or discussed in the main text.

The reviewer is likely referring to this statement in the methods: "For analyzing neural activity prior to lever press, we examined lever presses that occurred at least 2s following the previous lever press or random sound." This is the only mention of a 2s threshold in the methods. We now realize that the meaning of this statement, and exactly what analysis it refers to, was unfortunately not clear enough. The 2 second threshold referred to here was only used when looking at spontaneous activity prior to lever press, shown in Figure 3, not for any other analyses. The reason was that in Figure 3 we wanted to examine the motor preparatory activity of auditory cortical neurons up to 1s before a lever press; we therefore wanted to exclude the possibility that the initial time bins of this 1s window might contain residual responses to sounds or activity associated with lever presses occurring more than 1s before the lever press. For example, had

we selected lever presses that occurred at least 1s after the previous random/self-generated sound, the first time bins in Figure 3B could have been partly contaminated by responses to sounds occurring just before the 1s window shown in Figure 3. This is why we chose a 2s threshold. However, we realize that this rationale may not have been obvious from the description we provided in the methods and have therefore expanded it to make this more clear (see p. 26 in revised manuscript).

It might be useful to have an additional supplementary figure demonstrating that the key difference in response attenuation between genotypes is evident both for random sounds occurring >1 sec after the self-generated sound --- and, if possible, for random sounds occurring well before the lever press. (This may not be possible to show, I understand, because random-sound responses might be very noisy in the latter case when the animal's behaviour is less well controlled than during a lever press.)

We thank the reviewer for this excellent suggestion. It is indeed important to demonstrate that our key results are robust to changes in our analysis parameters, in this case specifically how sounds were selected for analysis. First, we wish to reiterate that, with one exception (see above), sounds were selected that occurred >1 sec following the previous sound (whether it was random or self-generated) and that this criterion was applied to both random and self-generated sounds. Following the suggestion of the reviewer, we have now repeated our key analysis comparing the modulation indices between the two genotypes (Figure 1G) using different temporal thresholds for the selection of sounds. Specifically, we repeated this analysis after selecting sounds that occurred at least 0.25, 0.5, 1 (as in Figure 1G), 1.5 and 2 seconds after the previous sound. These results are now shown in Supplementary Figure 2A. As this figure shows, regardless of the temporal criterion used there is a robust difference between the MI values of the two genotypes. Note that although the effect of sensory adaptation is expected to be much stronger at 0.25s than at 2s, it should affect responses to random and self-generated sounds equally, which is probably why the absolute MI values are relatively independent of the temporal threshold used.

The reviewer's suggestion of comparing the MI values of the two genotypes after selecting random sounds that precede a lever press is also a good one. Selecting such random sounds is likely more effective in minimizing differences in behavioral state between random and self-generated sounds. We therefore repeated the analysis shown in Figure 1G after selecting random sounds occurring between 0.05 and 1.05s before a lever press. The minimum time of 0.05s was necessary in order to prevent the responses to the random sound from being contaminated by the response to the subsequent self-generated sound and a total time window of 1s was used to obtain a sufficient number of random sounds for analysis. Although as expected this selection procedure reduced the number of random sounds selected for analysis considerably, they were still sufficiently numerous for calculating evoked responses. As shown in Supplementary Figure 2B of the revised manuscript, this selection procedure did not alter the difference in modulation indices between the two genotypes. Also, based on the reviewer's earlier comment, we repeated the analysis in Figure 1G after selecting random sounds that occurred when the lever was being held down, when somatosensory input is similar to that during self-generated sounds (Supplementary Figure 2C). In both cases, the difference in modulation indices between the

genotypes was similar to the original analysis in Figure 1G. We therefore conclude that our key finding of weaker attenuation of self-generated sounds in *Df(16)A^{+/-}* mice is robust to the specific criteria used for selecting random and self-generated sounds for analysis.

(2) The issue of hearing impairment in 22q11.2DS patients and at least some mouse models of 22q11.2DS is acknowledged in the Discussion, but discounted mostly on the grounds that it has not been demonstrated in all mouse models of 22q11.2DS and may be due to differences in laboratory conditions. True, but in fact the data in the manuscript provide a much stronger argument that hearing impairment would be an unlikely explanation for the effects observed here.

Hearing thresholds are essentially bimodal in both 22q11.2DS patients and the one mouse model of 22q11.2DS where they have been intensively investigated (i.e., some individuals have normal hearing and others have mild to moderate impairment). There is no evidence for bimodality in the *Df(16)A/+* data here. Moreover, the magnitude of evoked responses to random sounds appear to be statistically indistinguishable between the *Df(16)A/+* and WT mice, and there doesn't seem to be any obvious difference in the degree of self-generated sound attenuation depending on the magnitude of the response evoked by random sounds (i.e., there's no apparent nonlinearity in the Figure 1 E scatterplots), as might be expected if hearing impairment were a confound. Were spontaneous firing rates statistically indistinguishable in *Df(16)A/+* and WT mice (as suggested already by Figure 3 B)? If so then there's no evidence for a difference in baseline excitability of the cells either. With this degree of similarity between *Df(16)A/+* and WT mice in evoked responses to random sounds (and spontaneous rates?), it seems likely that the key conclusions regarding attenuation of self-generated sounds would be robust to any variability in hearing thresholds among *Df(16)A/+* mice even if it were present.

To be clear, I am not suggesting that the authors need to do additional analyses to address this point, only that they could make a stronger case in the Discussion that their conclusions are highly unlikely to have been confounded by hearing impairment, even if it does occur in a subset of the *Df(16)A/+* animals.

We thank the reviewer for these suggestions and have revised the discussion accordingly. We hasten to point out that our data can not directly address the question of whether hearing loss exists in the *Df(16)A^{+/-}* mice since this would require measuring hearing thresholds, for example using auditory brainstem responses, which we did not do. However, as the reviewer correctly points out, some of our observations would seem to argue against the presence of hearing loss in these mice. Based on our reading of the literature, it seems that alterations of the excitation/inhibition balance are frequently observed following peripheral hearing loss, which would be expected to manifest itself in altered responses to randomly generated sounds. However, responses to random sounds were not altered in the *Df(16)A^{+/-}* mice, and neither was any bimodality apparent in the responses distribution, which would be expected if only a subset of the mice had hearing loss, as previously reported (Zinnamon et al., 2022).

Regarding spontaneous rates, we only reported the average firing rates of putative pyramidal neurons and interneurons and did not find any difference between the genotypes (Figure 2C and D). However, these rates were calculated by averaging over the entire session and can therefore not be considered 'spontaneous' since they reflect at least partly the auditory-evoked responses. The reviewer suggests that the results shown in Figure 3B suggest that spontaneous rates are similar in the two genotypes. Although Figure 3B shows the normalized, rather than absolute firing rates, we did report in the text that at the beginning of the time period shown in this figure (-1000 to -900 ms), absolute firing rates did not differ between the genotypes (p. 8, line 1 in the revised manuscript). However, we have reservations about referring to these firing rates as 'spontaneous'. As Figure 3B shows, firing rates of auditory cortex neurons are strongly modulated by the animal's upcoming, and perhaps also ongoing, movement. This suggests that spontaneous rates should be calculated during periods of behavioral quiescence. Unfortunately, we were not able to unambiguously detect such periods and therefore do not feel confident making statements about spontaneous rates. However, since changes in excitation/inhibition balance that have been observed following hearing loss (Sanes, 2013) would be expected to affect average firing rates as well as responses to random sounds, the fact that these did not differ between the genotypes can be seen as arguing against the presence of hearing loss in *Df(16)A^{+/-}* mice. Finally, the fact that sensory adaptation and auditory thalamic inputs (i.e. number of thalamic neurons retrogradely labeled from cortex) were not affected in *Df(16)A^{+/-}* mice can be taken as further evidence against the possibility that the deficits in processing self-generated sounds in these mice arise from a more generalized auditory dysfunction. We have now added these points to the discussion (p. 12 of the revised manuscript).

Reviewer #2 (Remarks to the Author):

I enjoyed reading “Impaired corollary discharge signaling in the auditory cortex of a mouse model of schizophrenia predisposition” from the Sigurdsson lab. Even ancient philosophers knew that there had to be a mechanism such as the corollary discharge that enables animals to know whether sensations they are experiencing result from their own actions or from those of an approaching predator. Recently, clever investigators like these have begun to help the rest of us understand the neurobiology of this mechanism. This paper represents a step forward in the basic science of the corollary discharge, while also linking the dysfunction of this mechanism to a mouse model of schizophrenia related to the 22q.11 deletion.

I have very few suggestions to improve this paper, but here are a couple.

They say:

“Disruptions in the filtering of self-generated stimuli may also contribute to the symptoms of psychiatric disease.” They should temper this by saying instead: “Disruptions in the filtering of self-generated stimuli have been reported across the psychosis spectrum.”

We thank the reviewer for the helpful and constructive comments on the manuscript. We agree that “psychiatric disease” is too broad since these deficits have only been observed in schizophrenia and other psychotic disorders. We have therefore modified the sentence according to the reviewer’s suggestions.

They say:

“Such a failure to predict the sensory consequences of action could cause self-generated stimuli to be misattributed to an external source, leading to the hallucinations and delusions that are characteristic of schizophrenia (Feinberg and Guazzelli, 1999; Allen et al., 2007; Jones and Fernyhough, 2007; Fletcher and Frith, 2009; Thakkar et al., 2021).” Most, if not all, of these papers are reviews. They might consider adding another list of empirical papers that report a relationship between hallucinations/delusions and deficits in this system. They should search for Bonsal, Thakkar, Shergill, and Ford lab papers, among others.

We thank the reviewer for this helpful suggestion. We have now added a sentence in the introduction (p. 3) and in the discussion (p. 11) citing studies from these and other authors that have reported a correlation between deficits in predicting the sensory consequences of actions and the severity of hallucinations or delusions.

I am not aware of these type studies being done in humans with the 22q11.2 microdeletion. If there are some, the authors need to mention them. If there are not, they should mention that too.

This is a very good question. As far as we are aware, no study has examined the processing of self-generated stimuli (in the auditory or other sensory domains) in patients carrying the 22q11.2 microdeletion. That being said, the manifestation of schizophrenia in 22q11.2 carriers is similar to that of schizophrenia patients in general, in terms of clinical symptoms and structural brain abnormalities. Based on this, one would expect the deficits in attenuating self-generated sounds also to be present in 22q11.2 patients. But we acknowledge that this needs to be tested and have made a comment to this effect in the discussion of the revised manuscript (p. 12).

They cite the Singla et al Nature Neuroscience paper on a cerebellum-like circuit in the auditory system cancels responses to self-generated sounds. Can the authors say anything about the role of the cerebellum in this study? Did they not look in the cerebellum, or did they not see a signal in the cerebellum?

Examining the role of the cerebellum was beyond the scope of our study. Currently, there is not much known about how the cerebellum processes self-generated sounds in rodents (as opposed to the cerebellum-like dorsal cochlear nucleus examined in the Singla et al. paper) or how it responds to sounds in general. In contrast, our work and that of others had already examined and characterized the processing of self-generated sounds in the auditory cortex and there was already data suggesting what the underlying neuronal circuits could be (i.e. projections from the

motor cortex). We therefore felt we were in a better position to examine how self-generated sounds are processed in the auditory cortex of *Df(16)A^{+/-}* mice, rather than in the cerebellum. Another reason is that the auditory cortex is likely the source of the auditory responses that have been observed in patient studies. That being said, the reviewer's question regarding the possible role of the cerebellum is a very good one. Indeed, there are several human studies suggesting that the cerebellum plays a role in the attenuation of self-generated sensations (Blakemore et al. 1998; Knolle et al. 2013). We therefore believe that examining this role of the cerebellum in animal models will be an interesting direction for future research.

**Thank you for your excellent work,
Judith M. Ford**

Thank you Dr. Ford for the constructive and helpful review.

Reviewer #3 (Remarks to the Author):

This study investigates the distinction between self- and externally-generated stimuli in the Df16 mouse model of schizophrenia. The main finding is that Df16 vs WT mice show decreased attenuation of single neuron responses to self-generated sounds in auditory cortex. The researchers further show decreased preparatory motor activity in motor cortex and decreased numbers of motor-auditory projections in Df16 vs WT mice. They also establish that motor-auditory projections affect single neuron activity in auditory cortex in WT mice. The researchers conclude that decreased top-down signalling from motor cortex leads to a blurred distinction between self- and externally-generated stimuli in auditory cortex.

This is an elegant and original study that addresses the relevant question of how neural circuit dysfunction gives rise to sensory processing deficits relevant to schizophrenia. Reduced distinction between self and non-self has long been related to hallucinations and delusions in schizophrenia. This study now suggests a potential neural circuit mechanism for the well-established macro-level alterations in schizophrenia patients. The manuscript is well-written and the methodology is sound and described in sufficient detail to allow other researchers to reproduce the work. I have some concerns regarding the statistics and the interpretation of the results that I outline below. If adequately addressed, this work would make an extremely valuable contribution to both psychiatric and systems neuroscience.

We thank the reviewer for the detailed reading of the manuscript as well as the numerous helpful and constructive comments.

Comments:

1. In their statistical analyses, the researchers pool neurons across subjects. Although this practice has long been standard practice in the field, it is increasingly recognised that pooling dependent and independent observations violates the assumption of most statistical tests and can lead to biased statistical inferences (e.g., see here for a recent discussion <https://www.ncbi.nlm.nih.gov/pmc/articles/PMC7906290/>). Therefore, the authors might want to consider validating their key results with statistical tests that are designed to accurately deal with hierarchical data sets (e.g., linear mixed models, hierarchical bootstrap). At the very least, the researchers could show how consistent the results are across subjects (e.g., by presenting data grouped by subject).

We thank the reviewer for this suggestion. By pooling neurons across subjects we were following standard procedures in the field, but we acknowledge that this can lead to biased statistical estimates. We have therefore followed the reviewer's suggestion by repeating the statistical analysis of the key electrophysiological differences between the two genotypes (i.e. the attenuation of self-generated sounds and the motor-related signals) using the hierarchical bootstrapping method presented in the study cited by the reviewer (Saravanan et al., 2020). We have reported the p values of the hierarchical bootstrap next to the statistical analyses presented in the original manuscript and describe the hierarchical bootstrap procedure briefly in the methods (p. 28 of revised manuscript). In addition, for these comparisons (p. 5 and p. 8), we have also averaged values within each animal and report the mean \pm s.e.m. of the animal-averaged values for each genotype (which were similar to the averages across neurons) in the text as well as the p value of their genotype differences. Importantly, these additional statistical analyses confirm our key results.

Another of our key findings was that optogenetic stimulation of M2 terminals in Acx led to short-latency excitatory responses exclusively in the deeper cortical layers whereas inhibitory responses were more distributed across layers. Calculation of the statistical significance of these differences was based on the ratios of these responses in the upper and lower layers, pooled across neurons. Since the hierarchical bootstrap method is intended for comparisons of means rather than ratios, in this case we have taken the reviewer's alternative suggestion and computed the ratios of neurons showing excitatory and inhibitory responses in the upper and lower layers separately for each animal; this is now shown in Supplementary Figure 4A.

Finally, we note that neurons were pooled across subjects only for the electrophysiological analyses presented in Figures 1-4. For the analysis of retrogradely labeled neurons presented in Figures 5 and 6, we calculated for each animal the total number of labeled neurons in each brain region across all brain sections, resulting in one datapoint per animal; these data points are shown in Figures 5 and 6. Furthermore, all statistical comparisons between the two genotypes for each brain region were performed on these data points. Since these data points can be assumed to be independent, we have not changed the presentation and statistical analysis of the anatomical comparisons in the revised manuscript.

2. It remains open whether the findings of altered neural responses and anatomy have any functional relevance. The researchers did not assess whether the increased neural responses to self-generated stimuli were paralleled by changed behavioural responses. While this does not necessarily detract from the value of these findings, it seems a stretch to claim that the study establishes any “impaired” signalling, as this implies an impact on function. The authors could consider providing some additional results on the behavioral consequences of the altered neural responses. Alternatively, the authors could use more descriptive terms when talking about their results and discuss how the functional aspect could be considered in the future.

The reviewer is correct that the functional impact that the alterations we observed in the *Df(16)A^{+/-}* mice might have on the animals' behavior remains an open question. Our main goal in the study was to examine in these mice the same electrophysiological alterations that had previously been observed in schizophrenia patients, using a task that resembled as closely as possible those used in patient studies. It should be noted that in the patient studies reporting abnormalities in the processing of self-generated sounds, no behavioral correlates of these alterations were reported (although some studies have reported correlations with symptoms).

That being said, we believe that the question of how the reduced attenuation of neural responses to self-generated sounds in the *Df(16)A^{+/-}* mice might manifest itself at the behavioral level is highly relevant and addressing it is in fact one of our long-term research goals. One possibility is that *Df(16)A^{+/-}* mice might differ from their wild-type littermates in terms of their spontaneous behavioral reactions to self-generated sounds. In order to examine this possibility, for the revision we analyzed video recordings from a subset of the mice while they performed the lever-pressing task. As a first step, we examined the motion energy (absolute pixel change between frames) in the video signal corresponding to the face of the mice (see figure below, **A**) using FaceMap (<https://github.com/MouseLand/facemap>; Stringer et al., 2019) around the time a sound was self-generated. However, this did not reveal a behavioral response that could be unambiguously attributed to the sound itself, only gradual changes that might have reflected the motor act itself or reward anticipation/consumption; furthermore, no difference was observable between the genotypes. These results are shown in the figure below (**B**: motion energy is shown around the time of the lever press). We also considered the possibility that behavioral responses to the sounds manifest themselves in more specific components of the motion signal that might not have been visible in its average. We therefore used FaceMap to perform principal component (PC) analysis of the video signal from the face region and examined the scores of each PC (representing the magnitude of each component of the motion signal) around the onset of self-generated sounds. However, as with the average motion signal, none of the individual PCs appeared to reflect clear sound responses. Examples of the scores of the first 6 PCs are shown in the figure below (lower row) for one wild-type (**C**) and one *Df(16)A^{+/-}* (**D**) mouse.

Although our analysis does not suggest that differences in the processing of self-generated sounds between the genotypes are apparent in the spontaneous behavioral responses to these sounds, it is possible that such responses exist but are masked by other behaviors that accompany these sounds, such as lever pressing and reward consumption. Therefore, tasks designed specifically to examine the processing of self-generated sounds at the level of behavior and/or sensory processing are likely more appropriate. For example, one relatively straightforward prediction of our electrophysiological results is that in wild-type mice the intensity threshold for detecting sounds is higher when the sounds are self-generated than when they are randomly generated; furthermore, *Df(16)A^{+/-}* mice would be expected to have a lower threshold for detecting self-generated sounds compared to wild-type mice whereas the thresholds for detecting random sounds should be unaltered. One could also hypothesize that the ability of *Df(16)A^{+/-}* mice to distinguish sounds that are self-generated from those that are externally generated is impaired at the behavioral level. However, testing these and similar possibilities would first require developing and validating a behavioral task that could reveal differences in intensity thresholds between self-generated and random sounds in wild-type mice, or which could reveal their ability to discriminate between such sounds. We therefore feel that testing this and similar hypotheses is more of a long-term research goal and thus beyond the scope of the current revision. For now, in the revised manuscript (p. 11, line 33), we have made clear in the discussion that the behavioral impact of the electrophysiological alterations in the *Df(16)A^{+/-}* mice remains an open question and also suggest how this could be investigated, along the lines mentioned above.

We also acknowledge the reviewer's point regarding our use of 'impaired signaling' and agree that this phrase could be taken to imply a behavioral deficit, which we did not demonstrate (although 'impaired' is frequently used in the literature to describe purely electrophysiological alterations). We have therefore replaced 'impaired' with the more neutral adjective 'altered'

(denoting change), including in the title, or with more descriptive terms indicating the direction of the change (e.g. 'decreased').

3. Relatedly, the optogenetics experiments show that activity in motor cortex-auditory cortex projections has an impact on activity in auditory cortex. However, because these experiments are confined to wild-type animals, it remains open whether the motor-auditory projections function in the same way in Df16 and in WT mice, and whether activation of these projections during behaviour could even rescue the attenuation of responses to self-generated stimuli in auditory cortex. Again, this does not necessarily detract from the value of the study, as the authors already provide two neural circuit explanations for the observed auditory cortex responses (decreased preparatory activity in motor cortex AND decreased number of projections). However, it seems misleading to claim “weakened” projections when the functional strength of these projections has not been assessed in Df16 mice, and the authors could consider using more descriptive words when presenting and discussing their results.

We thank the reviewer for pointing this out. All other things being equal, a reduced number of M2 neurons projecting to Acx should also cause a corresponding decrease in the overall strength of the M2-Acx projection in the *Df(16)A^{+/-}* mice. Nonetheless, the reviewer is correct that the functional strength of these projections would need to be tested directly in order to definitively make this claim. We did in fact attempt to test this for the revision using the *in vivo* optogenetic approach described in Figure 4. Unfortunately, however, the variability in ChR2 expression between animals was unexpectedly high, and due to low numbers of available *Df(16)A^{+/-}* mice from our breeding colony, we did not foresee that we could continue with and complete these experiments in a timely fashion. We are therefore unable to make a statement about the functional strength of the M2-Acx projection in the *Df(16)A^{+/-}* mice. We also came to the conclusion that this question would be better addressed *in vitro* (Nelson et al., 2013), since this would allow the strength of synaptic currents to be directly examined and thus reveal synaptic mechanisms underlying an observed reduction synaptic strength (in contrast, our *in vivo* approach can only measure suprathreshold spiking responses). Although our lab does not have the ability to perform *in vitro* recordings, it is something we plan to examine in the future through collaborations. For now, we have therefore followed the reviewer's suggestion and used more descriptive words when describing the anatomical findings in Figure 6. We now say that these inputs are 'fewer' or 'decreased' in the *Df(16)A^{+/-}* mice, which more accurately reflects the findings (i.e. fewer M2 neurons projecting to the auditory cortex in these mice). We thank the reviewer for bringing our attention to this point and hope that we have addressed it satisfactorily.

4. The researchers show that light delivery to AudCx with ChR2-expressing M2 terminals leads to responses in some neurons. Can the researchers be sure that these responses are indeed mediated by the activation of M2 terminals, or might the observed responses be a direct effect of the light on the neurons? It seems that the experiments using light delivery to M2 (Fig. S2) provide some support for the first interpretation. The authors might want to consider being more explicit about this and/or consider additional arguments for their interpretation of the optogenetics results.

The reviewer raises a very important concern; light can indeed affect neuronal activity in the absence of ChR2 expression. By “a direct effect of the light on the neurons”, the reviewer is likely referring to light-induced heating of brain tissue, which has been reported to increase or decrease neuronal firing rates, depending on the brain region (Stujenske et al., 2015; Owen et al., 2019). Given that the light-induced changes in temperature and firing rate are relatively gradual (e.g. Owen et al., 2019, Figure 2b) we think it is unlikely that they can account for the short-latency excitatory responses we saw in deep-layer Acx neurons, although they could have influenced inhibitory responses that were quantified over longer timescales.

As the reviewer suggests, the fact that we obtained similar results (in terms of ratios and laminar distribution of excitatory and inhibitory responses) when delivering light directly to M2 cell bodies argues against the possibility that our results are due to light directly affecting auditory cortical neurons (for example via heating). However, it remains possible that in these recordings temperature-induced firing rate changes occurred in M2 cell bodies, which in turn influenced activity of downstream auditory cortical neurons. Furthermore, due to partial light leakage from the optic fiber and the patch cord, it is also possible that the light was visible to the mice and induced sensory responses. This is relevant, because auditory cortical neurons can respond to visual stimuli, in particular in the deeper layers, although these responses have been reported to occur at latencies beyond our analysis window of 50 ms (Morrill and Hasenstaub, 2018). Finally, light can induce photoelectric artefacts on the recording electrode, which could be detected as artefactual spikes or interfere with spike detection.

For the above reasons, we felt it was best to address the issue of opsin-independent light effects on neural activity experimentally. We therefore performed recordings in mice that had not been transfected with ChR2 in M2 and delivered the light to the auditory cortex with the same intensity and duration as in the experiments on ChR2-expressing mice. We then quantified the average firing rates surrounding light onset in these mice and performed the same analysis in ChR2-expressing mice for comparison. These results are now shown in Supplementary Figure 4E and F and described in the results (p. 9 of revised manuscript). Whereas in the ChR2-expressing mice, a brief excitatory short-latency response and a longer-latency and more long lasting inhibition is observed, no such responses are observed in the mice without ChR2 expression. Indeed, applying the same criteria for defining excited and inhibited neurons as we had used for the ChR2-expressing mice (described in the Methods) out of 161 neurons recorded in the mice without ChR2 expression we did not observe any excited neurons and only one inhibited neuron. We therefore conclude that the effects on Acx neuronal activity when delivering light to the Acx in

mice expressing ChR2 in M2 neurons reflects the activation of axonal projections from M2 and Acx.

The reviewer may also have noticed the slight decrease in firing rate in the 5 ms bin immediately preceding light onset in both groups, as well as a similar slight decrease in the mice without ChR2 expression in the first 5 ms bin after light onset. This is likely a consequence of the photoelectric artefact, which occurs at light onset. Such an artefact can corrupt the waveform of spikes occurring immediately before or after light onset and thus exclude them from being clustered together with other spikes of a neuron, thus leading to an artefactual decrease in firing rate. However, the 5 ms bins used in Supplementary Figures 4E and F exaggerate the temporal duration of this artefact; plotting the data with smaller bins shows that it is in fact mostly restricted to 1 ms before and after light onset. To illustrate this, in the figure below we have plotted for the reviewer the firing rates in 1 ms bins 10 ms before and after light onset in mice with (left) and without (right) ChR2 expression. The figure for the ChR2-expressing mice also shows that the excitatory responses in ChR2-expressing mice begin 3-4 ms after light onset and thus are unlikely to have been affected by the light artefact (also note that since the duration of the light pulses was 100 ms, the light artefact at light offset did not affect the responses presented here).

Minor suggestions:

Page 5, line 27 “and suggests that this learning is impaired in *Df(16)A*^{+/-} mice” -> This conclusion does not seem justified as attenuation is decreased both at early and late timepoints and there is no interaction between genotype and timepoint which would suggest a difference in learning.

The reviewer is indeed correct, an interaction would be necessary to demonstrate a learning impairment. We are also hesitant to claim normal learning in *Df(16)A*^{+/-} mice based on the lack of an interaction since the number of trials per block we used were rather large and a more fine-grained analysis might have been able to reveal a learning impairment (we tried including fewer trials per block, but this resulted in responses that were too noisy). We have therefore removed

the statement about learning impairments and reorganized the sentences slightly in order to emphasize the learning effect overall.

Page 5, line 30 “reveal a cellular basis for similar deficits seen in schizophrenia patients” -> This conclusion does not seem justified by the data, as this is a study in a mouse model of a genetic syndrome associated with a high risk for schizophrenia without any patients tested. The authors might want to consider using some modifiers (“could”, “potential”, etc.).

We agree with the reviewer and thank him/her for pointing this out. We have added “possible” before “cellular basis” in the revised manuscript.

Page 7, line 1: What test does the p-value refer to?

The p-value is from the rank-sum test comparing the modulation indices between *Df(16)A^{+/-}* and wildtype mice at the 900-1100 micrometer bin. We have now made this more clear in the revised version.

Figures captions: How many mice were used in each plot?

We thank the reviewer for pointing this out. In the Methods of the original manuscript we specified only the total number of *Df(16)A^{+/-}* and wild-type mice used in the study. We have now added information about how many mice were used in the electrophysiology, optogenetic and anatomical experiments to the methods. In the original manuscript, we did mention in the results the total number of mice from which neurons were recorded (p. 4 and 8 in the revised manuscript) and in which retrograde tracing was performed (p. 10 in the revised manuscript). We have now also included the number of mice from which the data in each subplot comes from at the end of the figure legends. Note that for a few of the electrophysiology analyses, data were not included from all animals which was due to neurons from some animals not meeting our selection criteria, which were based on their auditory responsiveness, as-described in the methods.

Figure 1/3/6: “Impaired” and “weaker” imply an impact to function, but the figures depict decreases in neuron activity/numbers which might or might not have an impact on function. The researcher could consider using more descriptive terms. This might seem like semantics but words can matter when presenting results related to a condition that is affecting people who might read or hear about this work.

This is a valid point. We have changed the titles of these figures accordingly using more descriptive terms: “Decreased attenuation ...” (Figure 1); “Diminished motor preparatory activity ...” (Figure 3); “Decreased motor cortical inputs ...” (Figure 6). We have also replaced ‘impaired’ and ‘weaker’ elsewhere in the manuscript in response to the reviewer’s main comments #2 and 3. We thank the reviewer for bringing this to our attention.

Figure 5, caption: “Percentages are relative to the total number of neurons in each brain region” -> “Percentages are relative to the total number of labelled neurons in each brain region” (I assume as the numbers seem to add up to 100).

We thank the reviewer for spotting this. Yes, the percentages shown are of the total number of retrogradely labeled neurons. We have added this to the figure legend.

References

- Morrill RJ, Hasenstaub AR (2018) Visual information present in infragranular layers of mouse auditory cortex. *J Neurosci* Available at: <http://dx.doi.org/10.1523/JNEUROSCI.3102-17.2018>.
- Nelson A, Schneider DM, Takatoh J, Sakurai K, Wang F, Mooney R (2013) A circuit for motor cortical modulation of auditory cortical activity. *J Neurosci* 33:14342–14353.
- Owen SF, Liu MH, Kreitzer AC (2019) Thermal constraints on in vivo optogenetic manipulations. *Nat Neurosci* Available at: <https://doi.org/10.1038/s41593-019-0422-3>.
- Rummell BP, Klee JL, Sigurdsson T (2016) Attenuation of Responses to Self-Generated Sounds in Auditory Cortical Neurons. *J Neurosci* 36:12010–12026.
- Sanes DH (2013) Synaptic and Cellular Consequences of Hearing Loss. In: *Deafness* (Kral A, Popper AN, Fay RR, eds), pp 129–149. New York, NY: Springer New York.
- Saravanan V, Berman GJ, Sober SJ (2020) Application of the hierarchical bootstrap to multi-level data in neuroscience. *Neuron Behav Data Anal Theory* 3 Available at: <https://www.ncbi.nlm.nih.gov/pubmed/33644783>.
- Schneider DM, Nelson A, Mooney R (2014) A synaptic and circuit basis for corollary discharge in the auditory cortex. *Nature* 513:189–194.
- Stringer C, Pachitariu M, Steinmetz N, Reddy CB, Carandini M, Harris KD (2019) Spontaneous behaviors drive multidimensional, brainwide activity. *Science* 364:255.
- Stujenske JM, Spellman T, Gordon JA (2015) Modeling the Spatiotemporal Dynamics of Light and Heat Propagation for In Vivo Optogenetics. *Cell Rep* 12:525–534.
- Wehr M, Zador AM (2005) Synaptic mechanisms of forward suppression in rat auditory cortex. *Neuron* 47:437–445.
- Zhou M, Liang F, Xiong XR, Li L, Li H, Xiao Z, Tao HW, Zhang LI (2014) Scaling down of balanced excitation and inhibition by active behavioral states in auditory cortex. *Nat Neurosci* 17:841–850.
- Zinnamon FA, Harrison FG, Wenas SS, Liu Q, Wang KH, Linden JF (2022) Increased central auditory gain and decreased parvalbumin-positive cortical interneuron density in the Df1/+ mouse model of schizophrenia correlate with hearing impairment. *Biological Psychiatry*

Global Open Science Available at:
<https://www.sciencedirect.com/science/article/pii/S2667174322000325>.

REVIEWERS' COMMENTS

Reviewer #1 (Remarks to the Author):

This is one of the more rigorous and thorough replies to reviews that I have had the pleasure to read. Thank you to the authors for the very comprehensive and careful consideration of questions raised. Points that I had misunderstood from the original text have been well clarified, and additional supplementary analyses (e.g. testing effects of varying the minimum separation between sounds, and testing sensory adaptation) very effectively address the questions from my original review. Indeed I am glad to see that the conclusions of the manuscript are further strengthened by the robustness of the results in these supplementary analyses.

Reviewer #3 (Remarks to the Author):

Congratulations to the authors on their thorough revision. All my concerns have been addressed, and I enjoyed reading the authors' exceptionally insightful rebuttal. I cannot wait to see where they will take this exciting line of work in the future!

REVIEWERS' COMMENTS

Reviewer #1 (Remarks to the Author):

This is one of the more rigorous and thorough replies to reviews that I have had the pleasure to read. Thank you to the authors for the very comprehensive and careful consideration of questions raised. Points that I had misunderstood from the original text have been well clarified, and additional supplementary analyses (e.g. testing effects of varying the minimum separation between sounds, and testing sensory adaptation) very effectively address the questions from my original review. Indeed I am glad to see that the conclusions of the manuscript are further strengthened by the robustness of the results in these supplementary analyses.

Reviewer #3 (Remarks to the Author):

Congratulations to the authors on their thorough revision. All my concerns have been addressed, and I enjoyed reading the authors' exceptionally insightful rebuttal. I cannot wait to see where they will take this exciting line of work in the future!

Author's response: We thank the reviewers once again for their constructive comments which greatly helped us to improve our manuscript.